# Q-learning with Nearest Neighbors

**Devavrat Shah** *
Massachusetts Institute of Technology
devavrat@mit.edu

**Qiaomin Xie** *
Massachusetts Institute of Technology
qxie@mit.edu

## Abstract

We consider model-free reinforcement learning for infinite-horizon discounted Markov Decision Processes (MDPs) with a *continuous* state space and unknown transition kernel, when only a single sample path under an arbitrary policy of the system is available. We consider the *Nearest Neighbor Q-Learning* (NNQL) algorithm to learn the optimal Q function using nearest neighbor regression method. As the main contribution, we provide tight finite sample analysis of the convergence rate. In particular, for MDPs with a $d$-dimensional state space and the discounted factor $\gamma \in (0,1)$, given an arbitrary sample path with "covering time" $L$, we establish that the algorithm is guaranteed to output an $\varepsilon$-accurate estimate of the optimal Q-function using $\widetilde{O}\big(L/(\varepsilon^3(1-\gamma)^7)\big)$ samples. For instance, for a well-behaved MDP, the covering time of the sample path under the purely random policy scales as $\widetilde{O}\big(1/\varepsilon^d\big)$, so the sample complexity scales as $\widetilde{O}\big(1/\varepsilon^{d+3}\big)$. Indeed, we establish a lower bound that argues that the dependence of $\widetilde{\Omega}\big(1/\varepsilon^{d+2}\big)$ is necessary.

## 1   Introduction

Markov Decision Processes (MDPs) are natural models for a wide variety of sequential decision-making problems. It is well-known that the optimal control problem in MDPs can be solved, in principle, by standard algorithms such as value and policy iterations. These algorithms, however, are often not directly applicable to many practical MDP problems for several reasons. First, they do not scale computationally as their complexity grows quickly with the size of the state space and especially for continuous state space. Second, in problems with complicated dynamics, the transition kernel of the underlying MDP is often unknown, or an accurate model thereof is lacking. To circumvent these difficulties, many model-free Reinforcement Learning (RL) algorithms have been proposed, in which one estimates the relevant quantities of the MDPs (e.g., the value functions or the optimal policies) from observed data generated by simulating the MDP.

A popular model-free Reinforcement Learning (RL) algorithm is the so called Q-learning [47], which directly learns the optimal action-value function (or Q function) from the observations of the system trajectories. A major advantage of Q-learning is that it can be implemented in an online, incremental fashion, in the sense that Q-learning can be run as data is being sequentially collected from the system operated/simulated under some policy, and continuously refines its estimates as new observations become available. The behaviors of standard Q-learning in *finite* state-action problems have by now been reasonably understood; in particular, both asymptotic and finite-sample convergence guarantees have been established [43, 22, 41, 18].

In this paper, we consider the general setting with *continuous* state spaces. For such problems, existing algorithms typically make use of a parametric function approximation method, such as a linear approximation [27], to learn a compact representation of the action-value function. In many of

the recently popularized applications of Q-learning, much more expressive function approximation method such as deep neural networks have been utilized. Such approaches have enjoyed recent empirical success in game playing and robotics problems [38, 29, 14]. Parametric approaches typically require careful selection of approximation method and parametrization (e.g., the architecture of neural networks). Further, rigorous convergence guarantees of Q-learning with deep neural networks are relatively less understood. In comparison, non-parametric approaches are, by design, more flexible and versatile. However, in the context of model-free RL with continuous state spaces, the convergence behaviors and finite-sample analysis of non-parametric approaches are less understood.

**Summary of results.** In this work, we consider a natural combination of the Q-learning with Kernel-based nearest neighbor regression for continuous state-space MDP problems, denoted as Nearest-Neighbor based Q-Learning (NNQL). As the main result, we provide *finite sample analysis* of NNQL for a *single, arbitrary* sequence of data for any infinite-horizon discounted-reward MDPs with continuous state space. In particular, we show that the algorithm outputs an $\varepsilon$-accurate (with respect to supremum norm) estimate of the optimal Q-function with high probability using a number of observations that depends polynomially on $\varepsilon$, the model parameters and the "cover time" of the sequence of the data or trajectory of the data utilized. For example, if the data was sampled per a completely random policy, then our generic bound suggests that the number of samples would scale as $\widetilde{O}(1/\varepsilon^{d+3})$ where $d$ is the dimension of the state space. We establish effectively matching lower bound stating that for *any* policy to learn optimal $Q$ function within $\varepsilon$ approximation, the number of samples required must scale as $\widetilde{\Omega}(1/\varepsilon^{d+2})$. In that sense, our policy is *nearly* optimal.

Our analysis consists of viewing our algorithm as a special case of a general *biased* stochastic approximation procedure, for which we establish non-asymptotic convergence guarantees. Key to our analysis is a careful characterization of the bias effect induced by nearest-neighbor approximation of the population Bellman operator, as well as the statistical estimation error due to the variance of finite, dependent samples. Specifically, the resulting Bellman nearest neighbor operator allows us to connect the update rule of NNQL to a class of stochastic approximation algorithms, which have *biased* noisy updates. Note that traditional results from stochastic approximation rely on unbiased updates and asymptotic analysis [35, 43]. A key step in our analysis involves decomposing the update into two sub-updates, which bears some similarity to the technique used by [22]. Our results make improvement in characterizing the finite-sample convergence rates of the two sub-updates.

In summary, the salient features of our work are

- **Unknown system dynamics:** We assume that the transition kernel and reward function of the MDP is unknown. Consequently, we cannot exactly evaluate the expectation required in standard dynamic programming algorithms (e.g., value/policy iteration). Instead, we consider a sample-based approach which learns the optimal value functions/policies by directly observing data generated by the MDP.

- **Single sample path:** We are given a single, sequential samples obtained from the MDP operated under an arbitrary policy. This in particular means that the observations used for learning are *dependent*. Existing work often studies the easier settings where samples can be generated at will; that is, one can sample any number of (independent) transitions from any given state, or reset the system to any initial state. For example, *Parallel Sampling* in [23]. We do not assume such capabilities, but instead deal with the realistic, challenging setting with a single path.

- **Online computation:** We assume that data arrives sequentially rather than all at once. Estimates are updated in an online fashion upon observing each new sample. Moreover, as in standard Q-learning, our approach does not store old data. In particular, our approach differs from other batch methods, which need to wait for all data to be received before starting computation, and require multiple passes over the data. Therefore, our approach is space efficient, and hence can handle the data-rich scenario with a large, increasing number of samples.

- **Non-asymptotic, near optimal guarantees:** We characterize the finite-sample convergence rate of our algorithm; that is, how many samples are needed to achieve a given accuracy for estimating the optimal value function. Our analysis is *nearly* tight in that we establish a lower bound that nearly matches our generic upper bound specialized to setting when data is generated per random policy or more generally any policy with random exploration component to it.

While there is a large and growing literature on Reinforcement Learning for MDPs, to the best of our knowledge, ours is the first result on Q-learning that simultaneously has all of the above four features.

Table 1: Summary of relevant work. See Appendix A for details.

| Specific work | Method | Continuous state space | Unknown transition Kernel | Single sample path | Online update | Non-asymptotic guarantees |
|---|---|---|---|---|---|---|
| [10], [36], [37] | Finite-state approximation | Yes | No | No | Yes | Yes |
| [43], [22], [41] | Q-learning | No | Yes | Yes | Yes | No |
| [20], [3], [18] | Q-learning | No | Yes | Yes | Yes | Yes |
| [23] | Q-learning | No | Yes | No | Yes | Yes |
| [42],[28] | Q-learning | Yes | Yes | Yes | Yes | No |
| [33], [32] | Kernel-based approximation | Yes | Yes | No | No | No |
| [19] | Value/Policy iteration | No | Yes | No | No | Yes |
| [44] | Parameterized TD-learning | No | Yes | Yes | Yes | No |
| [12] | Parameterized TD-learning | No | Yes | No | Yes | Yes |
| [8] | Parameterized TD-learning | No | Yes | Yes | Yes | Yes |
| [9] | Non-parametric LP | No | Yes | No | No | Yes |
| [30] | Fitted value iteration | Yes | Yes | No | No | Yes |
| [1] | Fitted policy iteration | Yes | Yes | Yes | No | Yes |
| Our work | Q-learning | Yes | Yes | Yes | Yes | Yes |

We summarize comparison with relevant prior works in Table 1. Detailed discussion can be found in Appendix A.

## 2 Setup

In this section, we introduce necessary notations, definitions for the framework of Markov Decision Processes that will be used throughout the paper. We also precisely define the question of interest.

**Notation.** For a metric space $E$ endowed with metric $\rho$, we denote by $C(E)$ the set of all bounded and measurable functions on $E$. For each $f \in C(E)$, let $\|f\|_\infty := \sup_{x \in E} |f(x)|$ be the supremum norm, which turns $C(E)$ into a Banach space $B$. Let $\text{Lip}(E, M)$ denote the set of Lipschitz continuous functions on $E$ with Lipschitz bound $M$, i.e.,

$$\text{Lip}(E, M) = \{f \in C(E) \mid |f(x) - f(y)| \leq M\rho(x, y), \ \forall x, y \in E\}.$$

The indicator function is denoted by $\mathbb{1}\{\cdot\}$. For each integer $k \geq 0$, let $[k] \triangleq \{1, 2, \ldots, k\}$.

**Markov Decision Process.** We consider a general setting where an agent interacts with a stochastic environment. This interaction is modeled as a discrete-time discounted Markov decision process (MDP). An MDP is described by a five-tuple $(\mathcal{X}, \mathcal{A}, p, r, \gamma)$, where $\mathcal{X}$ and $\mathcal{A}$ are the state space and action space, respectively. We shall utilize $t \in \mathbb{N}$ to denote time. Let $x_t \in \mathcal{X}$ be state at time $t$. At time $t$, the action chosen is denoted as $a_t \in \mathcal{A}$. Then the state evolution is Markovian as per some transition probability kernel with density $p$ (with respect to the Lebesgue measure $\lambda$ on $\mathcal{X}$). That is,

$$\Pr(x_{t+1} \in B | x_t = x, a_t = a) = \int_B p(y|x, a)\lambda(dy) \tag{1}$$

for any measurable set $B \in \mathcal{X}$. The one-stage reward earned at time $t$ is a random variable $R_t$ with expectation $\mathbb{E}[R_t | x_t = x, a_t = a] = r(x, a)$, where $r : \mathcal{X} \times \mathcal{A} \to \mathbb{R}$ is the expected reward function. Finally, $\gamma \in (0, 1)$ is the discount factor and the overall reward of interest is $\sum_{t=0}^{\infty} \gamma^t R_t$ The goal is to maximize the expected value of this reward. Here we consider a distance function $\rho : \mathcal{X} \times \mathcal{X} \to \mathbb{R}_+$ so that $(\mathcal{X}, \rho)$ forms a metric space. For the ease of exposition, we use $\mathcal{Z}$ for the joint state-action space $\mathcal{X} \times \mathcal{A}$.

We start with the following standard assumptions on the MDP:

**Assumption 1** (MDP Regularity)**.** *We assume that: (A1.) The continuous state space $\mathcal{X}$ is a compact subset of $\mathbb{R}^d$; (A2.) $\mathcal{A}$ is a finite set of cardinality $|\mathcal{A}|$; (A3.) The one-stage reward $R_t$ is non-negative and uniformly bounded by $R_{\max}$, i.e., $0 \leq R_t \leq R_{\max}$ almost surely. For each $a \in \mathcal{A}$, $r(\cdot, a) \in \text{Lip}(\mathcal{X}, M_r)$ for some $M_r > 0$. (A4.) The transition probability kernel $p$ satisfies*

$$|p(y|x, a) - p(y|x', a)| \leq W_p(y)\rho(x, x'), \qquad \forall a \in \mathcal{A}, \forall x, x', y \in \mathcal{X},$$

*where the function $W_p(\cdot)$ satisfies $\int_{\mathcal{X}} W_p(y)\lambda(dy) \leq M_p$.*

The first two assumptions state that the state space is compact and the action space is finite. The third and forth stipulate that the reward and transition kernel are Lipschitz continuous (as a function of the current state). Our Lipschitz assumptions are identical to (or less restricted than) those used in the work of [36], [11], and [17]. In general, this type of Lipschitz continuity assumptions are standard in the literature on MDPs with continuous state spaces; see, e.g., the work of [15, 16], and [6].

A Markov policy $\pi(\cdot|x)$ gives the probability of performing action $a \in \mathcal{A}$ given the current state $x$. A deterministic policy assigns each state a unique action. The *value* function for each state $x$ under policy $\pi$, denoted by $V^\pi(x)$, is defined as the expected discounted sum of rewards received following the policy $\pi$ from initial state $x$, i.e., $V^\pi(x) = \mathbb{E}_\pi\left[\sum_{t=0}^{\infty} \gamma^t R_t | x_0 = x\right]$. The *action-value function* $Q^\pi$ under policy $\pi$ is defined by $Q^\pi(x,a) = r(x,a) + \gamma \int_y p(y|x,a)V^\pi(y)\lambda(dy)$. The number $Q^\pi(x,a)$ is called the *Q-value* of the pair $(x,a)$, which is the return of initially performing action $a$ at state $s$ and then following policy $\pi$. Define

$$\beta \triangleq 1/(1-\gamma) \qquad \text{and} \qquad V_{\max} \triangleq \beta R_{\max}.$$

Since all the rewards are bounded by $R_{\max}$, it is easy to see that the value function of every policy is bounded by $V_{\max}$ [18, 40]. The goal is to find an optimal policy $\pi^*$ that maximizes the value from any start state. The optimal value function $V^*$ is defined as $V^*(x) = V^{\pi^*}(x) = \sup_\pi V^\pi(x)$, $\forall x \in \mathcal{X}$. The optimal action-value function is defined as $Q^*(x,a) = Q^{\pi^*}(x,a) = \sup_\pi Q^\pi(x,a)$. The Bellman optimality operator $F$ is defined as

$$(FQ)(x,a) = r(x,a) + \gamma \mathbb{E}\left[\max_{b \in \mathcal{A}} Q(x',b) \mid x,a\right] = r(x,a) + \gamma \int_{\mathcal{X}} p(y|x,a)\max_{b \in \mathcal{A}} Q(y,b)\lambda(dy).$$

It is well known that $F$ is a contraction with factor $\gamma$ on the Banach space $C(\mathcal{Z})$ [7, Chap. 1]. The optimal action-value function $Q^*$ is the unique solution of the Bellman's equation $Q = FQ$ in $C(\mathcal{X} \times \mathcal{A})$. In fact, under our setting, it can be show that $Q^*$ is bounded and Lipschitz. This is stated below and established in Appendix B.

**Lemma 1.** *Under Assumption 1, the function $Q^*$ satisfies that $\|Q^*\|_\infty \leq V_{\max}$ and that $Q^*(\cdot,a) \in \text{Lip}(\mathcal{X}, M_r + \gamma V_{\max} M_p)$ for each $a \in \mathcal{A}$.*

## 3 Reinforcement Learning Using Nearest Neighbors

In this section, we present the nearest-neighbor-based reinforcement learning algorithm. The algorithm is based on constructing a finite-state discretization of the original MDP, and combining Q-learning with nearest neighbor regression to estimate the $Q$-values over the discretized state space, which is then interpolated and extended to the original continuous state space. In what follows, we shall first describe several building blocks for the algorithm in Sections 3.1–3.4, and then summarize the algorithm in Section 3.5.

### 3.1 State Space Discretization

Let $h > 0$ be a pre-specified scalar parameter. Since the state space $\mathcal{X}$ is compact, one can find a finite set $\mathcal{X}_h \triangleq \{c_i\}_{i=1}^{N_h}$ of points in $\mathcal{X}$ such that

$$\min_{i \in [N_h]} \rho(x,c_i) < h, \ \forall x \in \mathcal{X}.$$

The finite grid $\mathcal{X}_h$ is called an $h$-net of $\mathcal{X}$, and its cardinality $n \equiv N_h$ can be chosen to be the $h$-covering number of the metric space $(\mathcal{X}, \rho)$. Define $\mathcal{Z}_h = \mathcal{X}_h \times \mathcal{A}$. Throughout this paper, we denote by $\mathcal{B}_i$ the ball centered at $c_i$ with radius $h$; that is, $\mathcal{B}_i \triangleq \{x \in \mathcal{X} : \rho(x,c_i) \leq h\}$.

### 3.2 Nearest Neighbor Regression

Suppose that we are given estimated Q-values for the finite subset of states $\mathcal{X}_h = \{c_i\}_{i=1}^n$, denoted by $q = \{q(c_i,a), c_i \in \mathcal{X}_h, a \in \mathcal{A}\}$. For each state-action pair $(x,a) \in \mathcal{X} \times \mathcal{A}$, we can predict its Q-value via a regression method. We focus on nonparametric regression operators that can be written as nearest neighbors averaging in terms of the data $q$ of the form

$$(\Gamma_{\text{NN}}q)(x,a) = \sum_{i=1}^n K(x,c_i)q(c_i,a), \qquad \forall x \in \mathcal{X}, a \in \mathcal{A}, \tag{2}$$

where $K(x,c_i) \geq 0$ is a weighting kernel function satisfying $\sum_{i=1}^n K(x,c_i) = 1, \forall x \in \mathcal{X}$. Equation (2) defines the so-called Nearest Neighbor (NN) operator $\Gamma_{\text{NN}}$, which maps the space $C(\mathcal{X}_h \times \mathcal{A})$

into the set of all bounded function over $\mathcal{X} \times \mathcal{A}$. Intuitively, in (2) one assesses the Q-value of $(x, a)$ by looking at the training data where the action $a$ has been applied, and by averaging their values. It can be easily checked that the operator $\Gamma_{\text{NN}}$ is non-expansive in the following sense:

$$\|\Gamma_{\text{NN}}q - \Gamma_{\text{NN}}q'\|_\infty \leq \|q - q'\|_\infty, \qquad \forall q, q' \in C(\mathcal{X}_h \times \mathcal{A}). \tag{3}$$

This property will be crucially used for establishing our results. $K$ is assumed to satisfy

$$K(x, y) = 0 \text{ if } \rho(x, y) \geq h, \qquad \forall x \in \mathcal{X}, y \in \mathcal{X}_h, \tag{4}$$

where $h$ is the discretization parameter defined in Section 3.1.[2] This means that the values of states located in the neighborhood of $x$ are more influential in the averaging procedure (2). There are many possible choices for $K$. In Section C we describe three representative choices that correspond to $k$-Nearest Neighbor Regression, Fixed-Radius Near Neighbor Regression and Kernel Regression.

### 3.3 A Joint Bellman-NN Operator

Now, we define the *joint Bellman-NN (Nearest Neighbor) operator*. As will become clear subsequently, it is this operator that the algorithm aims to approximate, and hence it plays a crucial role in the subsequent analysis.

For a function $q : \mathcal{Z}_h \to \mathbb{R}$, we denote by $\tilde{Q} \triangleq (\Gamma_{\text{NN}}q)$ the nearest-neighbor average extension of $q$ to $\mathcal{Z}$; that is,

$$\tilde{Q}(x, a) = (\Gamma_{\text{NN}}q)(x, a), \quad \forall(x, a) \in \mathcal{Z}.$$

The joint Bellman-NN operator $G$ on $\mathbb{R}^{|\mathcal{Z}_h|}$ is defined by composing the original Bellman operator $F$ with the NN operator $\Gamma_{\text{NN}}$ and then restricting to $\mathcal{Z}_h$; that is, for each $(c_i, a) \in \mathcal{Z}_h$,

$$(Gq)(c_i, a) \triangleq (F\Gamma_{\text{NN}}q)(c_i, a) = (F\tilde{Q})(c_i, a) = r(c_i, a) + \gamma \mathbb{E}\left[\max_{b \in \mathcal{A}}(\Gamma_{\text{NN}}q)(x', b) \mid c_i, a\right]. \tag{5}$$

It can be shown that $G$ is a contraction operator with modulus $\gamma$ mapping $\mathbb{R}^{|\mathcal{Z}_h|}$ to itself, thus admitting a unique fixed point, denoted by $q_h^*$; see Appendix E.2.

### 3.4 Covering Time of Discretized MDP

As detailed in Section 3.5 to follow, our algorithm uses data generated by an abritrary policy $\pi$ for the purpose of learning. The goal of our approach is to estimate the Q-values of *every* state. For there to be any hope to learn something about the value of a given state, this state (or its neighbors) must be visited at least once. Therefore, to study the convergence rate of the algorithm, we need a way to quantify how often $\pi$ samples from different regions of the state-action space $\mathcal{Z} = \mathcal{X} \times \mathcal{A}$.

Following the approach taken by [18] and [3], we introduce the notion of the *covering time* of MDP under a policy $\pi$. This notion is particularly suitable for our setting as our algorithm is based on *asynchronous* Q-learning (that is, we are given a single, sequential trajectory of the MDP, where at each time step one state-action pair is observed and updated), and the policy $\pi$ may be non-stationary. In our continuous state space setting, the covering time is defined with respect to the discretized space $\mathcal{Z}_h$, as follows:

**Definition 1** (Covering time of discretized MDP)**.** *For each $1 \leq i \leq n = N_h$ and $a \in \mathcal{A}$, a ball-action pair $(\mathcal{B}_i, a)$ is said to be visited at time $t$ if $x_t \in \mathcal{B}_i$ and $a_t = a$. The discretized state-action space $\mathcal{Z}_h$ is covered by the policy $\pi$ if all the ball-action pairs are visited at least once under the policy $\pi$. Define $\tau_{\pi,h}(x, t)$, the covering time of the MDP under the policy $\pi$, as the minimum number of steps required to visit all ball-action pairs starting from state $x \in \mathcal{X}$ at time-step $t \geq 0$. Formally, $\tau_{\pi,h}(x, t)$ is defined as*

$$\min\left\{s \geq 0 : x_t = x, \ \forall i \leq N_h, a \in \mathcal{A}, \ \exists t_{i,a} \in [t, t+s], \ \text{such that } x_{t_{i,a}} \in B_i \ \text{and } a_{t_{i,a}} = a, \ \text{under } \pi\right\},$$

*with notation that minimum over empty set is $\infty$.*

We shall assume that there exists a policy $\pi$ with bounded expected cover time, which guarantees that, asymptotically, all the ball-action pairs are visited infinitely many times under the policy $\pi$.

**Assumption 2.** *There exists an integer $L_h < \infty$ such that $\mathbb{E}[\tau_{\pi,h}(x,t)] \le L_h, \forall x \in \mathcal{X}, t > 0$. Here the expectation is defined with respect to randomness introduced by Markov kernel of MDP as well as the policy $\pi$.*

In general, the covering time can be large in the worst case. In fact, even with a finite state space, it is easy to find examples where the covering time is exponential in the number of states for every policy. For instance, consider an MDP with states $1, 2, \ldots, N$, where at any state $i$, the chain is reset to state 1 with probability $1/2$ regardless of the action taken. Then, every policy takes exponential time to reach state $N$ starting from state 1, leading to an exponential covering time.

To avoid the such bad cases, some additional assumptions are needed to ensure that the MDP is well-behaved. For such MDPs, there are a variety of polices that have a small covering time. Below we focus on a class of MDPs satisfying a form of the uniform ergodic assumptions, and show that the standard $\varepsilon$-greedy policy (which includes the purely random policy as special case by setting $\varepsilon = 1$) has a small covering time. This is done in the following two Propositions. Proofs can be found in Appendix D.

**Proposition 1.** *Suppose that the MDP satisfies the following: there exists a probability measure $\nu$ on $\mathcal{X}$, a number $\varphi > 0$ and an integer $m \ge 1$ such that for all $x \in \mathcal{X}$, all $t \ge 0$ and all policies $\mu$,*

$$\Pr_{\mu}(x_{m+t} \in \cdot \mid x_t = x) \ge \varphi \nu(\cdot). \tag{6}$$

*Let $\nu_{\min} \triangleq \min_{i \in [n]} \nu(\mathcal{B}_i)$, where we recall that $n \equiv N_h = |\mathcal{X}_h|$ is the cardinality of the discretized state space. Then the expected covering time of $\varepsilon$-greedy is upper bounded by $L_h = O\left(\frac{m|\mathcal{A}|}{\varepsilon \varphi \nu_{\min}} \log(n|\mathcal{A}|)\right)$.*

**Proposition 2.** *Suppose that the MDP satisfies the following: there exists a probability measure $\nu$ on $\mathcal{X}$, a number $\varphi > 0$ and an integer $m \ge 1$ such that for all $x \in \mathcal{X}$, all $t \ge 0$, there exists a sequence of actions $\hat{\mathbf{a}}(x) = (\hat{a}_1, \ldots, \hat{a}_m) \in \mathcal{A}^m$,*

$$\Pr(x_{m+t} \in \cdot \mid x_t = x, a_t = \hat{a}_1, \ldots, a_{t+m-1} = \hat{a}_m) \ge \varphi \nu(\cdot). \tag{7}$$

*Let $\nu_{\min} \triangleq \min_{i \in [n]} \nu(\mathcal{B}_i)$, where we recall that $n \equiv N_h = |\mathcal{X}_h|$ is the cardinality of the discretized state space. Then the expected covering time of $\varepsilon$-greedy is upper bounded by $L_h = O\left(\frac{m|\mathcal{A}|^{m+1}}{\varepsilon^{m+1} \varphi \nu_{\min}} \log(n|\mathcal{A}|)\right)$.*

### 3.5 Q-learning using Nearest Neighbor

We describe the nearest-neighbor Q-learning (NNQL) policy. Like Q-learning, it is a model-free policy for solving MDP. Unlike standard Q-learning, it is (relatively) efficient to implement as it does not require learning the Q function over entire space $\mathcal{X} \times \mathcal{A}$. Instead, we utilize the nearest neighbor regressed Q function using the learned Q values restricted to $\mathcal{Z}_h$. The policy assumes access to an existing policy $\pi$ (which is sometimes called the "exploration policy", and need not have any optimality properties) that is used to sample data points for learning.

The pseudo-code of NNQL is described in Policy 1. At each time step $t$, action $a_t$ is performed from state $Y_t$ as per the given (potentially non-optimal) policy $\pi$, and the next state $Y_{t+1}$ is generated according to $p(\cdot|Y_t, a_t)$. Note that the sequence of observed states $(Y_t)$ take continuous values in the state space $\mathcal{X}$.

The policy runs over *iteration* with each iteration lasting for a number of time steps. Let $k$ denote iteration count, $T_k$ denote time when iteration $k$ starts for $k \in \mathbb{N}$. Initially, $k = 0$, $T_0 = 0$, and for $t \in [T_k, T_{k+1})$, the policy is in iteration $k$. The iteration is updated from $k$ to $k + 1$ when starting with $t = T_k$, all ball-action $(\mathcal{B}_i, a)$ pairs have been visited at least once. That is, $T_{k+1} = T_k + \tau_{\pi,h}(Y_{T_k}, T_k)$. In the policy description, the counter $N_k(c_i, a)$ records how many times the ball-action pair $(\mathcal{B}_i, a)$ has been visited from the beginning of iteration $k$ till the current time $t$; that is, $N_k(c_i, a) = \sum_{s=T_k}^{t} \mathbb{1}\{Y_s \in \mathcal{B}_i, a_s = a\}$. By definition, the iteration $k$ ends at the first time step for which $\min_{(c_i, a)} N_k(c_i, a) > 0$.

During each iteration, the policy keeps track of the Q-function over the finite set $\mathcal{Z}_h$. Specifically, let $q^k$ denote the approximate Q-values on $\mathcal{Z}_h$ within iteration $k$. The policy also maintains $G^k q^k(c_i, a_t)$, which is a *biased* empirical estimate of the joint Bellman-NN operator $G$ applied to the estimates $q^k$.

---

**Policy 1** Nearest-Neighbor Q-learning

---

**Input**: Exploration policy $\pi$, discount factor $\gamma$, number of steps $T$, bandwidth parameter $h$, and initial state $Y_0$.

Construct discretized state space $\mathcal{X}_h$; initialize $t = k = 0$, $\alpha_0 = 1$, $q^0 \equiv 0$;

**Foreach** $(c_i, a) \in \mathcal{Z}_h$, set $N_0(c_i, a) = 0$; **end**

**repeat**

    Draw action $a_t \sim \pi(\cdot|Y_t)$ and observe reward $R_t$; generate the next state $Y_{t+1} \sim p(\cdot|Y_t, a_t)$;

    **Foreach** $i$ **such that** $Y_t \in \mathcal{B}_i$ **do**

        $\eta_N = \frac{1}{N_k(c_i, a_t)+1}$;

        **if** $N_k(c_i, a_t) > 0$ **then**

            $(G^k q^k)(c_i, a_t) = (1 - \eta_N)(G^k q^k)(c_i, a_t) + \eta_N \left(R_t + \gamma \max_{b \in \mathcal{A}}(\Gamma_{\text{NN}} q^k)(Y_{t+1}, b)\right)$;

        **else** $(G^k q^k)(c_i, a_t) = R_t + \gamma \max_{b \in \mathcal{A}}(\Gamma_{\text{NN}} q^k)(Y_{t+1}, b)$;

        **end**

        $N_k(c_i, a_t) = N_k(c_i, a_t) + 1$

    **end**

    **if** $\min_{(c_i, a) \in \mathcal{Z}_h} N_k(c_i, a) > 0$ **then**

        **Foreach** $(c_i, a) \in \mathcal{Z}_h$ **do**

            $q^{k+1}(c_i, a) = (1 - \alpha_k)q^k(c_i, a) + \alpha_k(G^k q^k)(c_i, a)$;

        **end**

        $k = k + 1$; $\alpha_k = \frac{\beta}{\beta + k}$;

        **Foreach** $(c_i, a) \in \mathcal{Z}_h$ **do** $N_k(c_i, a) = 0$; **end**

    **end**

    $t = t + 1$;

**until** $t \geq T$;

return $\hat{q} = q^k$

---

At each time step $t \in [T_k, T_{k+1})$ within iteration $k$, if the current state $Y_t$ falls in the ball $\mathcal{B}_i$, then the corresponding value $(G^k q^k)(c_i, a_t)$ is updated as

$$(G^k q^k)(c_i, a_t) = (1 - \eta_N)(G^k q^k)(c_i, a_t) + \eta_N \left(R_t + \gamma \max_{b \in \mathcal{A}}(\Gamma_{\text{NN}} q^k)(Y_{t+1}, b)\right), \tag{8}$$

where $\eta_N = \frac{1}{N_k(c_i, a_t)+1}$. We notice that the above update rule computes, in an incremental fashion, an estimate of the joint Bellman-NN operator $G$ applied to the current $q^k$ for each discretized state-action pair $(c_i, a)$, using observations $Y_t$ that fall into the neighborhood $\mathcal{B}_i$ of $c_i$. This nearest-neighbor approximation causes the estimate to be biased.

At the end of iteration $k$, i.e., at time step $t = T_{k+1} - 1$, a new $q^{k+1}$ is generated as follows: for each $(c_i, a) \in \mathcal{Z}_h$,

$$q^{k+1}(c_i, a) = (1 - \alpha_k)q^k(c_i, a) + \alpha_k(G^k q^k)(c_i, a). \tag{9}$$

At a high level, this update is similar to standard Q-learning updates — the Q-values are updated by taking a weighted average of $q^k$, the previous estimate, and $G^k q^k$, an one-step application of the Bellman operator estimated using newly observed data. There are two main differences from standard Q-learning: 1) the Q-value of each $(c_i, a)$ is estimated using all observations that lie *in its neighborhood* — a key ingredient of our approach; 2) we wait until all ball-action pairs are visited to update their Q-values, all at once.

Given the output $\hat{q}$ of Policy 1, we obtain an approximate Q-value for each (continuous) state-action pair $(x, a) \in \mathcal{Z}$ via the nearest-neighbor average operation, i.e., $Q_h^T(x, a) = (\Gamma_{\text{NN}} \hat{q})(x, a)$; here the superscript $T$ emphasizes that the algorithm is run for $T$ time steps with a sample size of $T$.

## 4   Main Results

As a main result of this paper, we obtain finite-sample analysis of NNQL policy. Specifically, we find that the NNQL policy converges to an $\varepsilon$-accurate estimate of the optimal $Q^*$ with time $T$ that has polynomial dependence on the model parameters. The proof can be found in Appendix E.

**Theorem 1.** *Suppose that Assumptions 1 and 2 hold. With notation $\beta = 1/(1-\gamma)$ and $C = M_r + \gamma V_{\max} M_p$, for a given $\varepsilon \in (0, 4V_{\max}\beta)$, define $h^* \equiv h^*(\varepsilon) = \frac{\varepsilon}{4\beta C}$. Let $N_{h^*}$ be the $h^*$-covering number of the metric space $(\mathcal{X}, \rho)$. For a universal constant $C_0 > 0$, after at most*

$$T = C_0 \frac{L_{h^*} V_{\max}^3 \beta^4}{\varepsilon^3} \log\left(\frac{2}{\delta}\right) \log\left(\frac{N_{h^*} |\mathcal{A}| V_{\max}^2 \beta^4}{\delta \varepsilon^2}\right)$$

*steps, with probability at least $1 - \delta$, we have $\left\| Q_{h^*}^T - Q^* \right\|_\infty \leq \varepsilon$.*

The theorem provides sufficient conditions for NNQL to achieve $\varepsilon$ accuracy (in sup norm) for estimating the optimal action-value function $Q^*$. The conditions involve the bandwidth parameter $h^*$ and the number of time steps $T$, both of which depend polynomially on the relevant problem parameters. Here an important parameter is the covering number $N_{h^*}$: it provides a measure of the "complexity" of the state space $\mathcal{X}$, replacing the role of the cardinality $|\mathcal{X}|$ in the context of discrete state spaces. For instance, for a unit volume ball in $\mathbb{R}^d$, the corresponding covering number $N_{h^*}$ scales as $O\big((1/h^*)^d\big)$ (cf. Proposition 4.2.12 in [46]). We take note of several remarks on the implications of the theorem.

**Sample complexity:** The number of time steps $T$, which also equals the number of samples needed, scales linearly with the covering time $L_{h^*}$ of the underlying policy $\pi$ to sample data for the given MDP. Note that $L_{h^*}$ depends implicitly on the complexities of the state and action space as measured by $N_{h^*}$ and $|\mathcal{A}|$. In the best scenario, $L_{h^*}$, and hence $T$ as well, is linear in $N_{h^*} \times |\mathcal{A}|$ (up to logarithmic factors), in which case we achieve (near) optimal linear sample complexity. The sample complexity $T$ also depends polynomially on the desired accuracy $\varepsilon^{-1}$ and the effective horizon $\beta = 1/(1-\gamma)$ of the discounted MDP — optimizing the exponents of the polynomial dependence remains interesting future work.

**Space complexity:** The space complexity of NNQL is $O(N_{h^*} \times |\mathcal{A}|)$, which is necessary for storing the values of $q^k$. Note that NNQL is a truly online algorithm, as each data point $(Y_t, a_t)$ is accessed only once upon observation and then discarded; no storage of them is needed.

**Computational complexity:** In terms of computational complexity, the algorithm needs to compute the NN operator $\Gamma_{\text{NN}}$ and maximization over $\mathcal{A}$ in each time step, as well as to update the values of $q^k$ for all $c_i \in \mathcal{X}_{h^*}$ and $a \in \mathcal{A}$ in each iteration. Therefore, the worst-case computational complexity per time step is $O(N_{h^*} \times |\mathcal{A}|)$, with an overall complexity of $O(T \times N_{h^*} \times |\mathcal{A}|)$. The computation can be potentially sped up by using more efficient data structures and algorithms for finding (approximate) nearest neighbors, such as k-d trees [5], random projection trees [13], Locality Sensitive Hashing [21] and boundary trees [26].

**Choice of $h^*$:** NNQL requires as input a user-specified parameter $h$, which determines the discretization granularity of the state space as well as the bandwidth of the (kernel) nearest neighbor regression. Theorem 1 provides a desired value $h^* = \varepsilon/4\beta C$, where we recall that $C$ is the Lipschitz parameter of the optimal action-value function $Q^*$ (see Lemma 1). Therefore, we need to use a small $h^*$ if we demand a small error $\varepsilon$, or if $Q^*$ fluctuates a lot with a large $C$.

## 4.1 Special Cases and Lower Bounds

Theorem 1, combined with Proposition 1, immediately yield the following bound that quantify the number of samples required to obtain an $\varepsilon$-optimal action-value function with high probability, if the sample path is generated per the uniformly random policy. The proof is given in Appendix F.

**Corollary 1.** *Suppose that Assumptions 1 and 2 hold, with $\mathcal{X} = [0,1]^d$. Assume that the MDP satisfies the following: there exists a uniform probability measure $\nu$ over $\mathcal{X}$, a number $\varphi > 0$ and an integer $m \geq 1$ such that for all $x \in \mathcal{X}$, all $t \geq 0$ and all policies $\mu$, $\Pr_\mu (x_{m+t} \in \cdot \mid x_t = x) \geq \varphi \nu(\cdot)$. After at most*

$$T = \kappa \frac{1}{\varepsilon^{d+3}} \log^3\left(\frac{1}{\delta\varepsilon}\right)$$

*steps, where $\kappa \equiv \kappa(|\mathcal{A}|, d, \beta, m)$ is a number independent of $\varepsilon$ and $\delta$, we have $\left\| Q_{h^*}^T - Q^* \right\|_\infty \leq \varepsilon$ with probability at least $1 - \delta$.*

Corollary 1 states that the sample complexity of NNQL scales as $\widetilde{O}\big(\frac{1}{\varepsilon^{d+3}}\big)$. We will show that this is effectively necessary by establishing a lower bound on *any* algorithm under *any* sampling policy! The proof of Theorem 2 can be found in Appendix G.

**Theorem 2.** *For any reinforcement learning algorithm $\hat{Q}_T$ and any number $\delta \in (0, 1)$, there exists an MDP problem and some number $T_\delta > 0$ such that*

$$\Pr\left[\left\|\hat{Q}_T - Q^*\right\|_\infty \geq C \left(\frac{\log T}{T}\right)^{\frac{1}{2+d}}\right] \geq \delta, \qquad \text{for all } T \geq T_\delta,$$

*where $C > 0$ is a constant. Consequently, for any reinforcement learning algorithm $\hat{Q}_T$ and any sufficiently small $\varepsilon > 0$, there exists an MDP problem such that in order to achieve*

$$\Pr\left[\left\|\hat{Q}_T - Q^*\right\|_\infty < \varepsilon\right] \geq 1 - \delta,$$

*one must have*

$$T \geq C'd \left(\frac{1}{\varepsilon}\right)^{2+d} \log\left(\frac{1}{\varepsilon}\right),$$

*where $C' > 0$ is a constant.*

## 5 Conclusions

In this paper, we considered the reinforcement learning problem for infinite-horizon discounted MDPs with a continuous state space. We focused on a reinforcement learning algorithm NNQL that is based on kernelized nearest neighbor regression. We established nearly tight finite-sample convergence guarantees showing that NNQL can accurately estimate optimal Q function using nearly optimal number of samples. In particular, our results state that the sample, space and computational complexities of NNQL scale polynomially (sometimes linearly) with the covering number of the state space, which is continuous and has uncountably infinite cardinality.

In this work, the sample complexity analysis with respect to the accuracy parameter is nearly optimal. But its dependence on the other problem parameters is not optimized. This will be an important direction for future work. It is also interesting to generalize approach to the setting of MDP beyond infinite horizon discounted problems, such as finite horizon or average-cost problems. Another possible direction for future work is to combine NNQL with a smart exploration policy, which may further improve the performance of NNQL. It would also be of much interest to investigate whether our approach, specifically the idea of using nearest neighbor regression, can be extended to handle infinite or even continuous action spaces.

## Acknowledgment

This work was supported in parts by NSF projects NeTs-1523546, TRIPODS-1740751, and CMMI-1462158.

## Footnotes

*Both authors are affiliated with Laboratory for Information and Decision Systems (LIDS). DS is with the Department of EECS as well as Statistics and Data Science Center at MIT.

[2]This assumption is not absolutely necessary, but is imposed to simplify subsequent analysis. In general, our results hold as long as $K(x, y)$ decays sufficiently fast with the distance $\rho(x, y)$.

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
