[Supplementary Material]

# A Related works

Given the large body of relevant literature, even surveying the work on Q-learning in a satisfactory manner is beyond the scope of this paper. Here we only mention the most relevant prior works, and compare them to ours in terms of the assumptions needed, the algorithmic approaches considered, and the performance guarantees provided. Table 1 provides key representative works from the literature and contrasts them with our result.

Q-learning has been studied extensively for finite-state MDPs. [43] and [22] are amongst the first to establish its asymptotic convergence. Both of them cast Q-learning as a stochastic approximation scheme — we utilize this abstraction as well. More recent work studies non-asymptotic performance of Q-learning; see, e.g., [41], [18], and [24]. Many variants of Q-learning have also been proposed and analyzed, including Double Q-learning [20], Speedy Q-learning [3], Phased Q-learning [23] and Delayed Q-learning [40].

A standard approach for continuous-state MDPs with known transition kernels, is to construct a reduced model by discretizing state space and show that the new finite MDP approximates the original one. For example, Chow and Tsitsiklis establish approximation guarantees for a multigrid algorithm when the state space is compact [10, 11]. This result is recently extended to average-cost problems and to general Borel state and action spaces in [37]. To reduce the computational complexity, Rust proposes a randomized version of the multigrid algorithm and provides a bound on its approximation accuracy [36]. Our approach bears some similarities to this line of work: we also use state space discretization, and impose similar continuity assumptions on the MDP model. However, we do not require the transition kernel to be known, nor do we construct a reduced model; rather, we learn the action-value function of the original MDP directly by observing its sample path.

The closest work to this paper is by Szepesvari and Smart [42], wherein they consider a variant of Q-learning combined with local function approximation methods. The algorithm approximates the optimal Q-values at a given set of sample points and interpolates it for each query point. Follow-up work considers combining Q-learning with linear function approximation [28]. Despite algorithmic similarity, their results are distinct from ours: they establish *asymptotic* convergence of the algorithm, based on the assumption that the data-sampling policy is stochastic stationary. In contrast, we provide finite-sample bounds, and our results apply for arbitrary sample paths (including non-stationary policies). Consequently, our analytical techniques are also different from theirs.

Some other closely related work is by Ormoneit and coauthors on model-free reinforcement learning for continuous state with unknown transition kernels [33, 32]. Their approach, called KBRL, constructs a kernel-based approximation of the conditional expectation that appears in the Bellman operator. Value iteration can then be run using the approximate Bellman operator, and asymptotic consistency is established for the resulting fixed points. A subsequent work demonstrates applicability of KBRL to practical large-scale problems [4]. Unlike our approach, KBRL is an offline, batch algorithm in which data is sampled at once and remains the same throughout the iterations of the algorithm. Moreover, the aforementioned work does not provide convergence rate or finite-sample performance guarantee for KBRL. The idea of approximating the Bellman operator by an empirical estimate, has also been used in the context of discrete state-space problems [19]. The approximate operator is used to develop Empirical Dynamic Programming (EDP) algorithms including value and policy iterations, for which non-asymptotic error bounds are provided. EDP is again an offline batch algorithm; moreover, it requires multiple, independent transitions to be sampled for each state, and hence does not apply to our setting with a single sample path.

In terms of theoretical results, most relevant is the work in [30], who also obtain finite-sample performance guarantees for continuous space problems with unknown transition kernels. Extension to the setting with a single sample path is considered in [1]. The algorithms considered therein, including fitted value iteration and Bellman-residual minimization based fitted policy iteration, are different from ours. In particular, these algorithms perform updates in a batch fashion and require storage of all the data throughout the iterations.

There are other papers that provide finite-sample guarantees, such as [25, 12]; however, their settings (availability of i.i.d. data), algorithms (TD learning) and proof techniques are very different from ours. The work by Bhandari et al. [8] also provides a finite sample analysis of TD learning with linear function approximation, for both the i.i.d. data model and a single trajectory. We also note that the work on PAC-MDP methods [34] explores the impact of exploration policy on

learning performance. The focus of our work is estimation of Q-functions rather than the problem of exploration; nevertheless, we believe it is an interesting future direction to study combining our algorithm with smart exploration strategies.

# B Proof of Lemma 1

*Proof.* Let $\mathcal{D}$ be the set of all functions $u : \mathcal{X} \times \mathcal{A} \to \mathbb{R}$ such that $\|u\|_\infty \leq V_{\max}$. Let $\mathcal{L}$ be the set of all functions $u : \mathcal{X} \times \mathcal{A} \to \mathbb{R}$ such that $u(\cdot, a) \in \mathrm{Lip}(\mathcal{X}, M_r + \gamma V_{\max} M_p), \forall a \in \mathcal{A}$. Take any $u \in \mathcal{D}$, and fix an arbitrary $a \in \mathcal{A}$. For any $x \in \mathcal{X}$, we have

$$|(Fu)(x,a)| = \left| r(x,a) + \gamma \mathbb{E}\left[\max_{b \in \mathcal{A}} u(Z,b)|x,a\right] \right| \leq R_{\max} + \gamma V_{\max} = V_{\max},$$

where the last equality follows from the definition of $V_{\max}$. This means that $Fu \in \mathcal{D}$. Also, for any $x, y \in \mathcal{X}$, we have

$$\begin{aligned}
|(Fu)(x,a) - (Fu)(y,a)| &= \left| r(x,a) - r(y,a) + \gamma \mathbb{E}\left[\max_{b \in \mathcal{A}} u(Z,b)|x,a\right] - \gamma \mathbb{E}\left[\max_{b \in \mathcal{A}} u(Z,b)|y,a\right] \right| \\
&\leq |r(x,a) - r(y,a)| + \gamma \left| \int_{\mathcal{X}} \max_{b \in \mathcal{A}} u(z,b)\left(p(z|x,a) - p(z|y,a)\right) \lambda(dz) \right| \\
&\leq M_r \rho(x,y) + \gamma \int_{\mathcal{X}} \left| \max_{b \in \mathcal{A}} u(z,b)\left(p(z|x,a) - p(z|y,a)\right) \right| \lambda(dz) \\
&\leq M_r \rho(x,y) + \gamma \|u\|_\infty \cdot \int_{\mathcal{X}} |p(z|x,a) - p(z|y,a)| \lambda(dz) \\
&\leq [M_r + \gamma V_{\max} M_p] \rho(x,y).
\end{aligned}$$

This means that $(Fu)(\cdot, a) \in \mathrm{Lip}(\mathcal{X}, M_r + \gamma V_{\max} M_p)$, so $Fu \in \mathcal{L}$. Putting together, we see that $F$ maps $\mathcal{D}$ to $\mathcal{D} \cap \mathcal{L}$, which in particular implies that $F$ maps $\mathcal{D} \cap \mathcal{L}$ to itself. Since $\mathcal{D} \cap \mathcal{L}$ is closed and $F$ is $\gamma$-contraction, both with respect to $\|\cdot\|_\infty$, the Banach fixed point theorem guarantees that $F$ has a unique fixed point $Q^* \in \mathcal{D} \cap \mathcal{L}$. This completes the proof of the lemma. □

# C Examples of Nearest Neighbor Regression Methods

Below we describe three representative nearest neighbor regression methods, each of which corresponds to a certain choice of the kernel function $K$ in the averaging procedure (2).

- *$k$-nearest neighbor ($k$-NN) regression*: For each $x \in \mathcal{X}$, we find its $k$ nearest neighbors in the subset $\mathcal{X}_h$ and average their Q-values, where $k \in [n]$ is a pre-specified number. Formally, let $c_{(i)}(x)$ denote the $i$-th closest data point to $x$ amongst the set $\mathcal{X}_h$. Thus, the distance of each state in $\mathcal{X}_h$ to $x$ satisfies $\rho(x, c_{(1)}(x)) \leq \rho(x, c_{(2)}(x)) \leq \cdots \leq \rho(x, c_{(n)}(x))$. Then the $k$-NN estimate for the Q-value of $(x,a)$ is given by $(\Gamma_{\mathrm{NN}}q)(x,a) = \frac{1}{k} \sum_{i=1}^{k} q(c_{(i)}(x), a)$. This corresponds to using in (2) the following weighting function

$$K(x, c_i) = \frac{1}{k} \mathbb{1}\left\{\rho(x, c_i) \leq \rho(x, c_{(k)}(x))\right\}.$$

  Under the definition of $\mathcal{X}_h$ in Section 3.1, the assumption (4) is satisfied if we use $k = 1$. For other values of $k$, the assumption holds with a potentially different value of $h$.

- **Fixed-radius near neighbor regression**: We find all neighbors of $x$ up to a threshold distance $h > 0$ and average their Q-values. The definition of $\mathcal{X}_h$ ensures that at least one point $c_i \in \mathcal{X}_h$ is within the threshold distance $h$, i.e., $\forall x \in \mathcal{X}, \exists c_i \in \mathcal{X}_h$ such that $\rho(x, c_i) \leq h$. We then can define the weighting function function according to

$$K(x, c_i) = \frac{\mathbb{1}\{\rho(x, c_i) \leq h\}}{\sum_{j=1}^{n} \mathbb{1}\{\rho(x, c_j) \leq h\}}.$$

- **Kernel regression**: Here the Q-values of the neighbors of $x$ are averaged in a *weighted* fashion according to some kernel function [31, 48]. The kernel function $\phi : \mathbb{R}^+ \to [0,1]$

takes as input a distance (normalized by the bandwidth parameter $h$) and outputs a similarity score between 0 and 1. Then the weighting function $K(x, c_i)$ is given by

$$K(x, c_i) = \frac{\phi\left(\frac{\rho(x,c_i)}{h}\right)}{\sum_{j=1}^{n} \phi\left(\frac{\rho(x,c_j)}{h}\right)}.$$

For example, a (truncated) Gaussian kernel corresponds to $\phi(s) = \exp\left(-\frac{s^2}{2}\right)\mathbb{1}\{s \leq 1\}$. Choosing $\phi(s) = \mathbb{1}\{s \leq 1\}$ reduces to the fixed-radius NN regression described above.

## D  Bounds on Covering time

### D.1  Proof of Proposition 1

*Proof.* Without loss of generality, we may assume that the balls $\{\mathcal{B}_i, i \in [n]\}$ are disjoint, since the covering time will only become smaller if they overlap with each other. Note that under $\varepsilon$-greedy policy, equation (6) implies that $\forall t \geq 0, \forall x \in \mathcal{X}, \forall a \in \mathcal{A}$,

$$\Pr\left(x_{m+t} \in \cdot, a_{m+t} = a | x_t = x\right) \geq \frac{\varepsilon}{|\mathcal{A}|}\psi\nu(\cdot). \tag{10}$$

First assume that the above assumption holds with $m = 1$. Let $M \triangleq n|\mathcal{A}|$ be the total number of ball-action pairs. Let $(\mathcal{P}_1, \ldots, \mathcal{P}_M)$ be a fixed ordering of these $M$ pairs. For each integer $t \geq 1$, let $K_t$ be the number of ball-action pairs visited up to time $t$. Let $T \triangleq \inf\{t \geq 1 : K_t = M\}$ be the first time when all ball-action pairs are visited. For each $k \in \{1, 2, \ldots, M\}$, let $T_k \triangleq \{t \geq 1 : K_t = k\}$ be the the first time when $k$ pairs are visited, and let $D_k \triangleq T_k - T_{k-1}$ be the time to visit the $k$-th pair after $k - 1$ pairs have been visited. We use the convention that $T_0 = D_0 = 0$. By definition, we have $T = \sum_{k=1}^{M} D_k$.

When $k - 1$ pairs have been visited, the probability of visiting a *new* pair is at least

$$\min_{I \subseteq [M], |I| = M-k+1} \Pr\left((x_{T_{k-1}+1}, a_{T_{k-1}+1}) \in \bigcup_{i \in I} \mathcal{P}_i | x_{T_{k-1}}\right)$$

$$= \min_{I \subseteq [M], |I| = M-k+1} \sum_{i \in I} \Pr\left((x_{T_{k-1}+1}, a_{T_{k-1}+1}) \in \mathcal{P}_i | x_{T_{k-1}}\right)$$

$$\geq (M - k + 1) \min_{i \in [M]} \Pr_\pi\left((x_{T_{k-1}+1}, a_{T_{k-1}+1}) \in \mathcal{P}_i | x_{T_{k-1}}\right)$$

$$\geq (M - k + 1) \cdot \varphi\nu_{\min} \cdot \frac{\varepsilon}{|\mathcal{A}|},$$

where the last inequality follows from Eq. (10). Therefore, $D_k$ is stochastically dominated by a geometric random variable with mean at most $\frac{|\mathcal{A}|}{(M-k+1)\varepsilon\varphi\nu_{\min}}$. It follows that

$$\mathbb{E}T = \sum_{k=1}^{M} \mathbb{E}D_k \leq \sum_{k=1}^{M} \frac{|\mathcal{A}|}{(M - k + 1)\varepsilon\varphi\nu_{\min}} = O\left(\frac{|\mathcal{A}|}{\varepsilon\varphi\nu_{\min}} \log M\right).$$

This prove the proposition for $m = 1$.

For general values of $m$, the proposition follows from a similar argument by considering the MDP only at times $t = m, 2m, 3m, \ldots$. □

### D.2  Proof of Proposition 2

*Proof.* We shall use a line of argument similar to that in the proof of Proposition 1. We assume that the balls $\{\mathcal{B}_i, i \in [n]\}$ are disjoint. Note that under $\varepsilon$-greedy policy $\pi$, for all $t \geq 0$, for all $x \in \mathcal{X}$, we have

$$\Pr_\pi\left(a_t = \hat{a}_1, \ldots, a_{t+m-1} = \hat{a}_m | x_t = x\right) \geq \left(\frac{\varepsilon}{|\mathcal{A}|}\right)^m. \tag{11}$$

The equation (7) implies that

$$\Pr_\pi\left(x_{t+m} \in \cdot | x_t = x\right)$$
$$\geq \Pr\left(x_{t+m} \in \cdot | x_t = x, a_t = \hat{a}_1, \ldots, a_{t+m-1} = \hat{a}_m\right) \times \Pr_\pi\left(a_t = \hat{a}_1, \ldots, a_{t+m-1} = \hat{a}_m | x_t = x\right)$$
$$\geq \psi\nu(\cdot)\left(\frac{\varepsilon}{|\mathcal{A}|}\right)^m.$$

Thus for each $a \in \mathcal{A}$,

$$\Pr_\pi\left(x_{t+m} \in \cdot, a_{t+m} = a | x_t = x\right) \geq \psi\nu(\cdot)\left(\tfrac{\varepsilon}{|\mathcal{A}|}\right)^{m+1}. \tag{12}$$

We first consider the case $m = 1$ and use the same notation as in the proof of Proposition 1. When $k-1$ pairs have been visited, the probability of visiting a *new* pair is at least

$$\min_{I \subseteq [M], |I| = M-k+1} \Pr\left((x_{T_{k-1}+1}, a_{T_{k-1}+1}) \in \bigcup_{i \in I} \mathcal{P}_i | x_{T_{k-1}}\right)$$
$$\geq (M-k+1) \min_{i \in [M]} \Pr_\pi\left((x_{T_{k-1}+1}, a_{T_{k-1}+1}) \in \mathcal{P}_i | x_{T_{k-1}}\right)$$
$$\geq (M-k+1) \cdot \varphi\nu_{\min} \cdot \left(\frac{\varepsilon}{|\mathcal{A}|}\right)^2,$$

where the last inequality follows from Eq. (12). Therefore, $D_k$, the time to visit the $k$-th pair after $k-1$ pairs have been visited, is stochastically dominated by a geometric random variable with mean at most $\frac{(|\mathcal{A}|/\varepsilon)^2}{(M-k+1)\varphi\nu_{\min}}$. It follows that

$$\mathbb{E}T = \sum_{k=1}^M \mathbb{E}D_k \leq \sum_{k=1}^M \frac{(|\mathcal{A}|/\varepsilon)^2}{(M-k+1)\varphi\nu_{\min}} = O\left(\frac{(|\mathcal{A}|/\varepsilon)^2}{\varphi\nu_{\min}} \log M\right).$$

This prove the proposition for $m = 1$.

For general values of $m$, the proposition follows from a similar argument by considering the MDP only at times $t = m, 2m, 3m, \ldots$. □

## E  Proof of the Main Result: Theorem 1

The proof of Theorem 1 consists of three key steps summarized as follows.

**Step 1. Stochastic Approximation.** Since the nearest-neighbor approximation of the Bellman operator induces a biased update for $q^k$ at each step, the key step in our proof is to analyze a Stochastic Approximation (SA) algorithm with *biased* noisy updates. In particular, we establish its finite-sample convergence rate in Theorem 3, which does not follow from available convergence theory. This result itself may be of independent interest.

**Step 2. Properties of NNQL.** To apply the stochastic approximation result to NNQL, we need to characterize some key properties of NNQL, including (i) the stability of the algorithm (i.e., the sequence $q^k$ stays bounded), as established in Lemma 3; (ii) the contraction property of the joint Bellman-NN operator, as established in Lemma 4; and (iii) the error bound induced by discretization of the state space, as established in Lemma 5.

**Step 3. Apply SA to NNQL.** We apply the stochastic approximation result to establish the finite-sample convergence of NNQL. In particular, step 2 above ensures that NNQL satisfies the assumptions in Theorem 3. Applying this theorem, we prove that NNQL converges to a neighborhood of $q_h^*$, the fixed point of the Joint Bellman-NN operator $G$, after a sufficient number of iterations. The proof of Theorem 1 is completed by relating $q_h^*$ to the true optimal Q-function $Q^*$, and by bounding the number of time steps in terms of the the number of iterations and the covering time.

### E.1  Stochastic Approximation

Consider a generic iterative stochastic approximation algorithm, where the iterative update rule is has the following form: let $\theta^t$ denote the *state* at time $t$, then it is updated as

$$\theta^{t+1} = \theta^t + \alpha_t\left(F(\theta^t) - \theta^t + w^{t+1}\right), \tag{13}$$

where $\alpha_t \in [0, 1]$ is a step-size parameter, $w^{t+1}$ is a noise term and $F$ is the functional update of interest.

**Theorem 3.** *Suppose that the mapping $F : \mathbb{R}^d \to \mathbb{R}^d$ has a unique fixed point $\theta^*$ with $\|\theta^*\|_\infty \leq V$, and is a $\gamma$-contraction with respect to the $\ell_\infty$ norm in the sense that*

$$\|F(\theta) - F(\theta')\|_\infty \leq \gamma \|\theta - \theta'\|_\infty$$

*for all $\theta, \theta' \in \mathbb{R}^d$, where $0 < \gamma < 1$. Let $\{\mathcal{F}^t\}$ be an increasing sequence of $\sigma$-fields so that $\alpha_t$ and $w^t$ are $\mathcal{F}^t$-measurable random variables, and $\theta^t$ be updated as per (13). Let $\delta_1, \delta_2, M, V$ be non-negative deterministic constants. Suppose that the following hold with probability 1:*

1. *The bias $\Delta^{t+1} = \mathbb{E}\left[w^{t+1} \,|\, \mathcal{F}^t\right]$ satisfies $\left\|\Delta^{t+1}\right\|_\infty \leq \delta_1 + \delta_2 \|\theta^t\|_\infty$, for all $t \geq 0$;*

2. *$\left\|w^{t+1} - \Delta^{t+1}\right\|_\infty \leq M$, for all $t \geq 0$;*

3. *$\|\theta^t\|_\infty \leq V$, for all $t \geq 0$.*

*Further, we choose*

$$\alpha_t = \frac{\beta}{\beta + t}, \tag{14}$$

*where $\beta = \frac{1}{1-\gamma}$. Then for each $0 < \varepsilon < \min\{2V\beta, 2M\beta^2\}$, after*

$$T = \frac{48VM^2\beta^4}{\varepsilon^3} \log\left(\frac{32dM^2\beta^4}{\delta\varepsilon^2}\right) + \frac{6V(\beta - 1)}{\varepsilon}$$

*iterations of (13), with probability at least $1 - \delta$, we have*

$$\left\|\theta^T - \theta^*\right\|_\infty \leq \beta(\delta_1 + \delta_2 V) + \varepsilon.$$

*Proof.* We define two auxiliary sequences: for $i \in [d]$, let $u_i^0 = \theta_i^0$, $r_i^0 = 0$ and

$$u_i^{t+1} = (1 - \alpha_t)u_i^t + \alpha_t \underbrace{(w_i^{t+1} - \Delta_i^{t+1})}_{\bar{w}_i^{t+1}},$$

$$r_i^{t+1} = \theta_i^{t+1} - u_i^{t+1}.$$

By construction, $\theta^t = u^t + r^t$ for all $t$. We first analyze the convergence rate of the $(u^t)$ sequence. One has

$$\begin{aligned}
u_i^{t+1} &= (1 - \alpha_t)u_i^t + \alpha_t \bar{w}_i^{t+1} \\
&= (1 - \alpha_t)(1 - \alpha_{t-1})u_i^{t-1} + (1 - \alpha_t)\alpha_{t-1}\bar{w}_i^t + \alpha_t \bar{w}_i^{t+1} \\
&= \sum_{j=1}^{t+1} \eta^{t+1,j} \bar{w}_i^j,
\end{aligned}$$

where we define

$$\eta^{t+1,j} := \alpha_{j-1} \cdot \prod_{l=j}^{t}(1 - \alpha_l).$$

Note that the centered noise $\bar{w}_i^{t+1} := w_i^{t+1} - \Delta_i^{t+1}$ satisfies

$$\begin{aligned}
\mathbb{E}\left[\bar{w}_i^{t+1} | \mathcal{F}^t\right] &= \mathbb{E}\left[w_i^{t+1} | \mathcal{F}^t\right] - \Delta_i^{t+1} = 0, \\
\mathbb{E}\left[|\bar{w}_i^{t+1}| \,|\, \mathcal{F}^t\right] &= \mathbb{E}\left[|w_i^{t+1} - \Delta_i^{t+1}| \,|\, \mathcal{F}^t\right] \leq M. \tag{15}
\end{aligned}$$

Now

$$\mathbb{E}\left[\eta^{t+1,j} \bar{w}_i^j \,|\, \mathcal{F}^{j-1}\right] = \eta^{t+1,j} \mathbb{E}\left[\bar{w}_i^j \,|\, \mathcal{F}^{j-1}\right] = 0. \tag{16}$$

With the linear learning rate defined in Eq. (14), and the fact that $\beta = \frac{1}{1-\gamma} > 1$, $\forall j \in [1, t+1]$, we have

$$\eta^{t+1,j} = \frac{\beta}{j - 1 + \beta} \cdot \prod_{l=j}^{t} \frac{l}{l + \beta} < \frac{\beta}{j} \prod_{l=j}^{t} \frac{l}{l + 1} = \frac{\beta}{t + 1}. \tag{17}$$

Since the centered noise sequence $\{\bar{w}_i^1, \bar{w}_i^2, \ldots, \bar{w}_i^{t+1}\}$ is uniformly bounded by $M > 0$, it follows that

$$\left| \eta^{t+1,j} \bar{w}_i^j \right| \leq \frac{M\beta}{t+1}. \tag{18}$$

Define, for $1 \leq s \leq t+1$,

$$z_s^{t+1,i} := \sum_{j=1}^{s} \eta^{t+1,j} \bar{w}_i^j, \tag{19}$$

and $z_0^{t+1,i} = 0$. Then it follows that

$$\mathbb{E}\left[ z_{s+1}^{t+1,i} | \mathcal{F}^s \right] = z_s^{t+1,i}. \tag{20}$$

And from (16)-(18), it follows that

$$|z_{s+1}^{t+1,i} - z_s^{t+1,i}| \leq \frac{M\beta}{t+1}. \tag{21}$$

That is, $z_s^{t+1,i}$ is a Martingale with bounded differences. And $u_i^{t+1} = z_{t+1}^{t+1,i}$. This, using Azuma-Hoeffding's inequality, will provide us desired bound on $|u_i^{t+1}|$. To that end, let us recall the Azuma-Hoeffding's inequality.

**Lemma 2** (Azuma-Hoeffding). *Let $X_j$ be Martingale with respect to filtration $\mathcal{F}_j$, i.e. $\mathbb{E}[X_{j+1}|\mathcal{F}_j] = X_j$ for $j \geq 1$ with $X_0 = 0$. Further, let $|X_j - X_{j-1}| \leq c_j$ with probability 1 for all $j \geq 1$. Then for all $\varepsilon \geq 0$,*

$$\Pr\left[ |X_n| \geq \varepsilon \right] \leq 2 \exp\left( -\frac{\varepsilon^2}{2 \sum_{j=1}^{n} c_j^2} \right).$$

Applying the lemma to $z_j^{t+1,i}$ for $j \geq 0$ with $z_0^{t+1,i} = 0$, (21) and the fact that $u_i^{t+1} = z_{t+1}^{t+1,i}$, we obtain that

$$\Pr\left( \left| u_i^{t+1} \right| > \varepsilon \right) \leq 2 \exp\left( -\frac{(t+1)\varepsilon^2}{2M^2\beta^2} \right). \tag{22}$$

Therefore, by union bound we obtain

$$\Pr\left( \exists t \geq T_1 \text{ such that } \left| u_i^t \right| > \varepsilon \right) \leq \sum_{t=T_1}^{\infty} \Pr\left( \left| u_i^t \right| > \varepsilon \right)$$

$$\leq 2 \sum_{t=T_1}^{\infty} \exp\left( -\frac{t\varepsilon^2}{2M^2\beta^2} \right)$$

$$= \frac{2 \exp\left( -\frac{T_1\varepsilon^2}{2M^2\beta^2} \right)}{1 - \exp\left( -\frac{\varepsilon^2}{2M^2\beta^2} \right)}$$

$$\leq \frac{8M^2\beta^2}{\varepsilon^2} \exp\left( -\frac{T_1\varepsilon^2}{2M^2\beta^2} \right),$$

where the last step follows from the fact that $e^{-x} \leq 1 - \frac{x}{2}$ for $0 \leq x \leq \frac{1}{2}$, and $\varepsilon \leq M\beta$. By a union bound over all $i \in [d]$, we deduce that

$$\Pr\left( \exists t \geq T_1 \text{ such that } \left\| u^t \right\|_\infty > \varepsilon \right) \leq \sum_{i \in [d]} \Pr\left( \exists t \geq T_1 \text{ such that } \left| u_i^t \right| > \varepsilon \right)$$

$$\leq \frac{8dM^2\beta^2}{\varepsilon^2} \exp\left( -\frac{T_1\varepsilon^2}{2M^2\beta^2} \right). \tag{23}$$

Next we focus on the residual sequence $(r^t)$. Assume that $\forall t \geq T_1$, $\|u^t\|_\infty \leq \varepsilon_1$, where $0 < \varepsilon_1 < \min\{V, M\beta\}$. For each $i \in [d]$ and $t \geq T_1$, we get

$$
\begin{aligned}
&\left|r_i^{t+1} - \theta_i^*\right| \\
&= \left|\theta_i^{t+1} - u_i^{t+1} - \theta_i^*\right| && \text{by definition} \\
&= \left|\theta_i^t + \alpha_t\left(F_i(u^t + r^t) - u_i^t - r_i^t + w_i^{t+1}\right) - u_i^t - \alpha_t\left(-u_i^t + w_i^{t+1} - \Delta_i^{t+1}\right) - \theta_i^*\right| && \text{by definition} \\
&= \left|r_i^t + \alpha_t\left(F_i(u^t + r^t) - r_i^t\right) - \theta_i^* + \alpha_t\Delta_i^{t+1}\right| && \text{rearranging} \\
&= \left|(1 - \alpha_t)(r_i^t - \theta_i^*) + \alpha_t\left(F_i(u^t + r^t) - \theta_i^*\right) + \alpha_t\Delta_i^{t+1}\right| && \text{rearranging} \\
&\leq (1 - \alpha_t)\left|r_i^t - \theta_i^*\right| + \alpha_t\gamma\left\|u^t + r^t - \theta^*\right\|_\infty + \alpha_t\left\|\Delta^{t+1}\right\|_\infty && F \text{ is } \gamma\text{-contraction} \\
&\leq (1 - \alpha_t)\left|r_i^t - \theta_i^*\right| + \alpha_t\gamma\left\|r^t - \theta^*\right\|_\infty + \alpha_t\gamma\varepsilon_1 + \alpha_t\left(\delta_1 + \delta_2\left\|\theta^t\right\|_\infty\right) && \|u^t\|_\infty \leq \varepsilon_1, \forall t \geq T_1 \\
&\leq (1 - \alpha_t)\left|r_i^t - \theta_i^*\right| + \alpha_t\gamma\left\|r^t - \theta^*\right\|_\infty + \alpha_t\left(\gamma\varepsilon_1 + \delta_1 + \delta_2 V\right) && \|\theta^t\|_\infty \leq V
\end{aligned}
$$

Taking the maximum over $i \in [d]$ on both sides, we obtain

$$
\begin{aligned}
\left\|r^{t+1} - \theta^*\right\|_\infty &\leq (1 - \alpha_t)\left\|r^t - \theta^*\right\|_\infty + \alpha_t\gamma\left\|r^t - \theta^*\right\|_\infty + \alpha_t\left(\gamma\varepsilon_1 + \delta_1 + \delta_2 V\right) \\
&= \Big(1 - \underbrace{(1 - \gamma)}_{\frac{1}{\beta}}\alpha_t\Big)\underbrace{\left\|r^t - \theta^*\right\|_\infty}_{D_t} + \alpha_t\underbrace{\left(\gamma\varepsilon_1 + \delta_1 + \delta_2 V\right)}_{H}, \qquad \forall t \geq T_1.
\end{aligned}
$$

For any $\varepsilon_2 > 0$, we will show that after at most

$$
T_2 \triangleq \frac{3V(T_1 + \beta - 1)}{\varepsilon_2}
$$

iterations, we have

$$
\left\|r^{T_2} - \theta^*\right\|_\infty \leq H\beta + \varepsilon_2.
$$

If for some $T \in [T_1, \infty)$ there holds $D_T \leq H\beta + \varepsilon_2$, then we have

$$
\begin{aligned}
D_{T+1} &\leq \left(1 - \frac{\alpha_t}{\beta}\right)(H\beta + \varepsilon_2) + \alpha_t H && \alpha_t \leq \beta \\
&\leq H\beta + \varepsilon_2
\end{aligned}
$$

Indeed by induction, we have

$$
D_t \leq H\beta + \varepsilon_2, \quad \forall t \geq T.
$$

Let $\widehat{T} \triangleq \sup\{t \geq T_1 : D_t > H\beta + \varepsilon_2\}$ be the last time that $D_t$ exceeds $H\beta + \varepsilon_2$. For each $T_1 \leq t \leq \widehat{T}$, the above argument implies that we must have $D_t > H\beta + \varepsilon_2$. We can rewrite the iteration for $D_{\widehat{T}}$ as follows:

$$
\begin{aligned}
D_{\widehat{T}} - H\beta &\leq \left(D_{\widehat{T}-1} - H\beta\right)\left(1 - \frac{\alpha_{\widehat{T}-1}}{\beta}\right) \\
&\leq (D_{T_1} - H\beta)\prod_{j=T_1}^{\widehat{T}-1}\left(1 - \frac{\alpha_j}{\beta}\right) && \text{Iteration, } D_t - H\beta > \varepsilon_2 > 0 \\
&= (D_{T_1} - H\beta)\frac{T_1 + \beta - 1}{\widehat{T} + \beta - 1} && \alpha_j = \frac{\beta}{j + \beta}.
\end{aligned}
$$

But we have the bound

$$
\begin{aligned}
D_{T_1} &= \|r^{T_1} - \theta^*\|_\infty \\
&= \|\theta^{T_1} - u^{T_1} - \theta^*\|_\infty \\
&\leq \|\theta^{T_1}\|_\infty + \|u^{T_1}\|_\infty + \|\theta^*\|_\infty \\
&\leq 3V,
\end{aligned}
$$

where the last step holds because $\|\theta^{T_1}\|_\infty \leq V$, $\|\theta^*\| \leq V$ and $\|u^{T_1}\| \leq \varepsilon_1 \leq V$ by assumption. It follows that

$$
D_{\widehat{T}} - H\beta \leq \frac{3V(T_1 + \beta - 1)}{\widehat{T}}.
$$

Using the fact that $\varepsilon_2 \leq D_{\widehat{T}} - H\beta$, we get that

$$\widehat{T} \leq T_2 = \frac{3V(T_1 + \beta - 1)}{\varepsilon_2}.$$

Therefore, for each $\varepsilon_2 > 0$, conditioned on the event

$$\left\{ \forall t \geq T_1, \left| u_i^t \right| \leq \varepsilon_1 \right\},$$

after at most $T_2$ iterations, we have

$$\left\| r^{T_2} - \theta^* \right\|_\infty \leq H\beta + \varepsilon_2.$$

It then follows from the relationship $\theta^t = u^t + r^t$ that

$$\left\| \theta^{T_2} - \theta^* \right\|_\infty \leq \left\| r^{T_2} - \theta^* \right\|_\infty + \left\| u^{T_2} \right\|_\infty \leq H\beta + \varepsilon_1 + \varepsilon_2 = \beta(\delta_1 + \delta_2 V) + \beta\varepsilon_1 + \varepsilon_2.$$

By (23), taking

$$\delta = \frac{8dM^2\beta^2}{\varepsilon_1^2} \exp\left( -\frac{T_1\varepsilon_1^2}{2M^2\beta^2} \right),$$

i.e.,

$$T_1 = \frac{2M^2\beta^2}{\varepsilon_1^2} \log\left( \frac{8dM^2\beta^2}{\delta\varepsilon_1^2} \right),$$

we are guaranteed that

$$\Pr\left( \forall t \geq T_1, \left\| u^t \right\|_\infty \leq \varepsilon_1 \right) \geq 1 - \delta.$$

By setting $\varepsilon_1 = \frac{\varepsilon}{2\beta} \leq \min\{V, M\beta\}$, and $T_2 = \frac{3V(T_1+\beta-1)}{\varepsilon_2}$, i.e.,

$$T_2 = \frac{48VM^2\beta^4}{\varepsilon^3} \log\left( \frac{32dM^2\beta^4}{\delta\varepsilon^2} \right) + \frac{6V(\beta-1)}{\varepsilon},$$

we obtain that the desire result. $\qquad\square$

## E.2 Properties of NNQL

We first introduce some notations. Let $\mathcal{Y}^k$ be the set of all samples drawn at iteration $k$ of the NNQL algorithms and $\mathcal{F}^k$ be the filtration generated by the sequence $\mathcal{Y}^0, \mathcal{Y}^1, \ldots, \mathcal{Y}^{k-1}$. Thus $\{\mathcal{F}^k\}$ is an increasing sequence of $\sigma$-fields. We denote by $\mathcal{Y}_k(c_i, a) = \{Y_t \in \mathcal{Y}_k | Y_t \in \mathcal{B}_i, a_t = a\}$ the set of observations $Y_t$ that fall into the neighborhood $\mathcal{B}_i$ of $c_i$ and with action $a_t = a$ at iteration $k$. Thus the biased estimator $G^k$ (8) for the joint Bellman-NN operator at the end of iteration $k$ can be written as

$$(G^k q)(c_i, a) = \frac{1}{|\mathcal{Y}_k(c_i, a)|} \sum_{Y_t \in \mathcal{Y}_k(c_i, a)} \left[ R_t + \gamma \max_{b \in \mathcal{A}}(\Gamma_{\text{NN}}q^k)(Y_{t+1}, b) \right].$$

The updater rule of NNQL (9) can be written as

$$q^{k+1}(c_i, a) = q^k(c_i, a) + \alpha_k \left[ (Gq^k)(c_i, a) - q^k(c_i, a) + w^{k+1}(c_i, a) \right],$$

where

$$\begin{aligned}
w^{k+1}(c_i, a) &= (G^k q^k)(c_i, a) - (Gq^k)(c_i, a) \\
&= \frac{1}{|\mathcal{Y}_k(c_i, a)|} \sum_{Y_t \in \mathcal{Y}_k(c_i, a)} \left[ R_t + \gamma \max_{b \in \mathcal{A}}(\Gamma_{\text{NN}}q^k)(Y_{t+1}, b) \right] \\
&\quad - r(c_i, a) - \gamma\mathbb{E}\left[ \max_{b \in \mathcal{A}}(\Gamma_{\text{NN}}q^k)(x', b) \mid c_i, a, \mathcal{F}^k \right].
\end{aligned}$$

### E.2.1 Stability of NNQL

We first show the stability of NNQL, which is summarized in the following Lemma.

**Lemma 3** (Stability of NNQL). *Assume that the immediate reward is uniformly bounded by $R_{\max}$ and define $\beta = \frac{1}{1-\gamma}$ and $V_{\max} = \beta R_{\max}$. If the initial action-value function $q^0$ is uniformly bounded by $V_{\max}$, then we have*

$$\left\|q^k\right\|_\infty \leq V_{\max}, \quad and \quad \left|w^{k+1}(c_i, a) - \mathbb{E}\left[w^{k+1}(c_i, a) \,|\, \mathcal{F}^k\right]\right| \leq 2V_{\max}, \qquad \forall k \geq 0.$$

*Proof.* We first prove that $\left\|q^k\right\|_\infty \leq V_{\max}$ by induction. For $k = 0$, it holds by the assumption. Now assume that for any $0 \leq \tau \leq k$, $\left\|q^\tau\right\|_\infty \leq V_{\max}$. Thus

$$\left|q^{k+1}(c_i, a)\right|$$
$$= \left|q^k(c_i, a) + \alpha_k\left[(G^k q^k)(c_i, a) - q^k(c_i, a)\right]\right|$$
$$= \left|(1 - \alpha_k)\, q^k(c_i, a) + \frac{\alpha_k}{|\mathcal{Y}_k(c_i, a)|} \sum_{Y_t \in \mathcal{Y}_k(c_i, a)}\left[R_t + \gamma \max_{b \in \mathcal{A}}(\Gamma_{\mathrm{NN}} q^k)(Y_{t+1}, b)\right]\right|$$
$$\leq \left|(1 - \alpha_k)\, q^k(c_i, a)\right| + \frac{\alpha_k}{|\mathcal{Y}_k(c_i, a)|} \sum_{Y_t \in \mathcal{Y}_k(c_i, a)}\left(|R_t| + \gamma\left|\max_{b \in \mathcal{A}}(\Gamma_{\mathrm{NN}} q^k)(Y_{t+1}, b)\right|\right)$$
$$= (1 - \alpha_k)\left|q^k(c_i, a)\right| + \frac{\alpha_k}{|\mathcal{Y}_k(c_i, a)|} \sum_{Y_t \in \mathcal{Y}_k(c_i, a)}\left(|R_t| + \gamma \max_{b \in \mathcal{A}}\left|\sum_{j=1}^n K(Y_{t+1}, c_j) q^k(c_j, b)\right|\right)$$
$$\leq (1 - \alpha_k) V_{\max} + \alpha_k\left(R_{\max} + \gamma \max_{b \in \mathcal{A}}\left|\sum_{j=1}^n K(Y_{t+1}, c_j)\right| V_{\max}\right)$$
$$= V_{\max},$$

where the last equality follows from the fact that $\sum_{j=1}^n K(Y_{t+1}, c_j) = 1$. Therefore, for all $k \geq 0$, $\left\|q^k\right\|_\infty \leq V_{\max}$. The bound on $w^{k+1}$ follows from

$$\left|w^{k+1}(c_i, a) - \mathbb{E}\left[w^{k+1}(c_i, a) \,|\, \mathcal{F}^k\right]\right|$$
$$= \left|(G^k q^k)(c_i, a) - (G q^k)(c_i, a) - \mathbb{E}\left[(G^k q^k)(c_i, a) - (G q^k)(c_i, a) \,|\, \mathcal{F}^k\right]\right|$$
$$= \left|(G^k q^k)(c_i, a) - \mathbb{E}\left[(G^k q^k)(c_i, a) \,|\, \mathcal{F}^k\right]\right|$$
$$= \left|\frac{1}{|\mathcal{Y}_k(c_i, a)|} \sum_{Y_t \in \mathcal{Y}_k(c_i, a)}\left[R_t + \gamma \max_{b \in \mathcal{A}}(\Gamma_{\mathrm{NN}} q^k)(Y_{t+1}, b)\right]\right.$$
$$\left.- \mathbb{E}\left[\frac{1}{|\mathcal{Y}_k(c_i, a)|} \sum_{Y_t \in \mathcal{Y}_k(c_i, a)}\left[R_t + \gamma \max_{b \in \mathcal{A}}(\Gamma_{\mathrm{NN}} q^k)(Y_{t+1}, b)\right] \,|\, \mathcal{F}^k\right]\right|$$
$$\leq 2R_{\max} + \frac{1}{|\mathcal{Y}_k(c_i, a)|} \sum_{Y_t \in \mathcal{Y}_k(c_i, a)} \gamma\left|\max_{b \in \mathcal{A}}\sum_{j=1}^n K(Y_{t+1}, c_j) q^k(c_j, b)\right|$$
$$+ \gamma\left|\mathbb{E}\left[\frac{1}{|\mathcal{Y}_k(c_i, a)|} \sum_{Y_t \in \mathcal{Y}_k(c_i, a)} \max_{b \in \mathcal{A}}\sum_{j=1}^n K(Y_{t+1}, c_j) q^k(c_j, b) \,|\, c_i, a, \mathcal{F}^k\right]\right|$$
$$\leq 2R_{\max} + 2\gamma V_{\max}$$
$$= 2V_{\max}.$$

$\square$

### E.2.2 A contraction operator

The following Lemma states that the joint Bellman-NN operator $G$ is a contraction with modulus $\gamma$, and has a unique fixed point that is bounded.

**Lemma 4** (Contraction of the Joint-Bellman-NN operator). *For each fixed $h > 0$, the operator $G$ defined in Eq. (5) is a contraction with modulus $\gamma$ with the supremum norm. There exists a unique function $q_h^*$ such that*

$$(Gq_h^*)(c_i, a) = q_h^*(c_i, a), \qquad \forall (c_i, a) \in \mathcal{Z}_h,$$

*where $\|q_h^*\|_\infty \leq V_{\max}$.*

*Proof.* Let $\mathcal{D}$ be the set of all functions $q : \mathcal{X}_h \times \mathcal{A} \to \mathbb{R}$ such that $\|q\|_\infty \leq V_{\max}$. We first show that the operator $G$ maps $\mathcal{D}$ into itself. Take any $q \in \mathcal{D}$, and fix an arbitrary $a \in \mathcal{A}$. For any $c_i \in \mathcal{X}_h$, we have

$$|(Gq)(c_i, a)| = \left| r(c_i, a) + \gamma \mathbb{E} \left[ \max_{b \in \mathcal{A}} (\Gamma_{\text{NN}} q)(x', b) | c_i, a \right] \right|$$

$$\leq |r(c_i, a)| + \gamma \left| \int_{\mathcal{X}} \left[ \max_{b \in \mathcal{A}} \left( \sum_{j=1}^{N_h} K(y, c_j) q(c_j, b) \right) \right] p(y|c_i a) \lambda(dy) \right|$$

$$\leq |r(c_i, a)| + \gamma \int_{\mathcal{X}} \left[ \max_{b \in \mathcal{A}} \sum_{j=1}^{N_h} K(y, c_j) |q(c_j, b)| \right] p(y|c_i a) \lambda(dy)$$

$$\leq R_{\max} + \gamma V_{\max}$$

$$= V_{\max},$$

where the last step follows from the definition of $V_{\max}$. This means that $Gq \in \mathcal{D}$, so $G$ maps $\mathcal{D}$ to itself.

Now, by the definition of $G$ in Eq. (5), $\forall q, q' \in \mathcal{D}$, we have

$$\|Gq - Gq'\|_\infty = \max_{i \in [n], a \in \mathcal{A}} |(Gq)(c_i, a) - (Gq')(c_i, a)|$$

$$\leq \gamma \max_{i \in [n], a \in \mathcal{A}} \left| \mathbb{E} \left[ \max_{b \in \mathcal{A}} \left( \sum_{j=1}^{n} K(x', c_j) (q(c_j, b) - q'(c_j, b)) \right) \mid c_i, a \right] \right|$$

$$\leq \gamma \max_{i \in [n], a \in \mathcal{A}} \mathbb{E} \left[ \max_{b \in \mathcal{A}} \left( \sum_{j=1}^{n} K(x', c_j) |q(c_j, b) - q'(c_j, b)| \right) \mid c_i, a \right]$$

$$\leq \gamma \max_{i \in [n], a \in \mathcal{A}} \mathbb{E} \left[ \max_{b \in \mathcal{A}} \left( \sum_{j=1}^{n} K(x', c_j) \|q - q'\|_\infty \right) \mid c_i, a \right]$$

$$\leq \gamma \|q - q'\|_\infty$$

Therefore $G$ is indeed a contraction on $\mathcal{D}$ with respect to the supremum norm. The Banach fixed point theorem guarantees that $G$ has a unique fixed point $q_h^* \in \mathcal{D}$. This completes the proof. $\qquad\square$

### E.2.3   Discretization error

For each $q \in C(\mathcal{Z}_h)$, we can obtain an extension to the original continuous state space via the Nearest Neighbor operator. That is, define

$$Q(x, a) = (\Gamma_{\text{NN}} q)(x, a), \forall (x, a) \in \mathcal{Z}.$$

The following lemma characterizes the distance between the optimal action-value function $Q^*$ and the extension of the fixed-point of the joint NN-Bellman operator $G$ to the space $\mathcal{Z}$.

**Lemma 5** (Discretization error). *Define*

$$Q_h^* = \Gamma_{NN} q_h^*.$$

*Let $Q^*$ be the optimal action-value function for the original MDP. Then we have*

$$\|Q_h^* - Q^*\| \leq \beta C h,$$

*where $C = M_r + \gamma V_{\max} M_p$ and $\beta = \frac{1}{1-\gamma}$.*

*Proof.* Consider an operator $H$ on $C(\mathcal{Z})$ defined as follows:

$$(HQ)(x,a) = (\Gamma_{\text{NN}}(FQ))(x,a)$$
$$= \sum_{i=1}^{n} K(x,c_i)\left\{ r(c_i,a) + \gamma \mathbb{E}\left[\max_{b\in\mathcal{A}} Q(x',b) \mid c_i,a\right]\right\} \tag{24}$$

We can show that $H$ is a contraction operator with modulus $\gamma$.

$$\|HQ_1 - HQ_2\|_\infty = \max_{a\in\mathcal{A}} \sup_{x\in\mathcal{X}} |(HQ_1)(x,a) - (HQ_2)(x,a)|$$

$$= \gamma \max_{a\in\mathcal{A}} \sup_{x\in\mathcal{X}} \left| \mathbb{E}\left[\max_{b\in\mathcal{A}}\left(\sum_{i=1}^{n} K(x,c_i)\left(Q_1(x',b) - Q_2(x',b)\right)\right) \mid c_i,a\right]\right|$$

$$\leq \gamma \max_{a\in\mathcal{A}} \sup_{x\in\mathcal{X}} \mathbb{E}\left[\max_{b\in\mathcal{A}}\left(\sum_{i=1}^{n} K(x,c_i)|Q_1(x',b) - Q_2(x',b)|\right) \mid c_i,a\right]$$

$$\leq \gamma \max_{a\in\mathcal{A}} \sup_{x\in\mathcal{X}} \mathbb{E}\left[\max_{b\in\mathcal{A}}\left(\sum_{i=1}^{n} K(x,c_i)\|Q_1 - Q_2\|_\infty\right) \mid c_i,a\right]$$

$$= \gamma \|Q_1 - Q_2\|_\infty$$

We can conclude that $H$ is a contraction operator mapping $C(\mathcal{Z})$ to $C(\mathcal{Z})$. Thus $H$ has a unique fixed point $\tilde{Q} \in C(\mathcal{Z})$. Note that

$$H(\Gamma_{\text{NN}}q) = \Gamma_{\text{NN}}(F(\Gamma_{\text{NN}}q)) = \Gamma_{\text{NN}}(Gq),$$

we thus have

$$HQ_h^* = H\left(\Gamma_{\text{NN}}q_h^*\right) = \Gamma_{\text{NN}}(Gq_h^*) = \Gamma_{\text{NN}}(q_h^*) = Q_h^*.$$

That is, the fixed point of $H$ is exactly the extension of the fixed point of $G$ to $\mathcal{Z}$. Therefore, we have

$$\|Q_h^* - Q^*\|_\infty = \|HQ_h^* - HQ^* + HQ^* - Q^*\|_\infty$$
$$\leq \|HQ_h^* - HQ^*\|_\infty + \|HQ^* - Q^*\|_\infty$$
$$\leq \gamma \|Q_h^* - Q^*\|_\infty + \|HQ^* - Q^*\|_\infty.$$

It follows that

$$\|Q_h^* - Q^*\|_\infty \leq \frac{1}{1-\gamma}\|HQ^* - Q^*\|_\infty$$

$$= \frac{1}{1-\gamma}\|\Gamma_{\text{NN}}(FQ^*) - Q^*\|_\infty$$

$$= \frac{1}{1-\gamma}\|\Gamma_{\text{NN}}(Q^*) - Q^*\|_\infty$$

$$= \frac{1}{1-\gamma} \sup_{x\in\mathcal{X}} \max_{a\in\mathcal{A}} \left|\sum_{i=1}^{n} K(x,c_i)Q^*(c_i,a) - Q^*(x,a)\right|$$

Recall that $Q^*(\cdot,a)$ is Lipschitz with parameter $C = M_r + \gamma V_{\max}M_p$ (see Lemma 1), i.e., for each $a \in \mathcal{A}$,

$$|Q^*(x,a) - Q^*(y,a)| \leq C\rho(x,y).$$

From the state space discretization step, we know that the finite grid $\{c_i\}_{i=1}^{N_h}$ is an $h$-net in $\mathcal{X}$. Therefore, for each $x \in \mathcal{X}$, there exists $c_i \in \mathcal{X}_h$ such that

$$\rho(x,c_i) \leq h.$$

Thus $\sum_{i=1}^{n} K(x,c_i) = 1$. Recall our assumption that the weighting function satisfies $K(x,y) = 0$ if $\rho(x,y) \geq h$. For each $a \in \mathcal{A}$, then we have

$$\sup_{x\in\mathcal{X}}\left|\sum_{i=1}^{n} K(x,c_i)Q^*(c_i,a) - Q^*(x,a)\right| = \sup_{x\in\mathcal{X}}\left|\sum_{c_i\in\mathcal{B}_{x,h}} K(x,c_i)Q^*(c_i,a) - Q^*(x,a)\right|$$

$$\leq \sup_{x\in\mathcal{X}} \sum_{c_i\in\mathcal{B}_{x,h}} K(x,c_i)|Q^*(c_i,a) - Q^*(x,a)|$$

$$\leq Ch$$

This completes the proof. $\qquad\square$

### E.3 Applying the Stochastic Approximation Theorem to NNQL

We first apply Theorem 3 to establish that NNQL converges to a neighborhood of $q_h^*$, the fixed point of the Joint Bellman-NN operator $G$, after a sufficiently large number of iterations. This is summarized in the following theorem.

**Theorem 4.** *Let Assumptions 1 and 2 hold. Then for each $0 < \varepsilon < 2V_{\max}\beta$, after*

$$k = \frac{192V_{\max}^3\beta^4}{\varepsilon^3}\log\left(\frac{128dV_{\max}^2\beta^4}{\delta\varepsilon^2}\right) + \frac{4V_{\max}(\beta-1)}{\varepsilon}$$

*iterations of Nearest-Neighbor Q-learning, with probability at least $1 - \delta$, we have*

$$\left\|q^k - q_h^*\right\|_\infty \leq \beta(\delta_1 + \delta_2 V_{\max}) + \varepsilon.$$

*Proof.* We will show that NNQL satisfies the assumptions of Theorem 3. It follows from Lemma 4 that the operator $G$ is a $\gamma$-contraction with a unique fixed point $\|q_h^*\|_\infty \leq V_{\max}$. For each $Y_t \in \mathcal{Y}_k(c_i, a)$, we have $\rho(Y_t, c_i) \leq h$, $a_t = a$. Thus

$$\left|\mathbb{E}\left[w^{k+1}(c_i, a)|\mathcal{F}^k\right]\right|$$
$$= \left|\mathbb{E}\left[\frac{1}{|\mathcal{Y}_k(c_i, a)|}\sum_{Y_t \in \mathcal{Y}_k(c_i, a)}\left[R_t + \gamma\max_{b \in \mathcal{A}}(\Gamma_{\text{NN}}q^k)(Y_{t+1}, b)\right] \mid \mathcal{F}^k\right]\right.$$
$$\left. - r(c_i, a) - \gamma\mathbb{E}\left[\max_{b \in \mathcal{A}}(\Gamma_{\text{NN}}q^k)(x', b) \mid c_i, a, \mathcal{F}^k\right]\right|$$
$$\leq \left|\mathbb{E}\left[\frac{1}{|\mathcal{Y}_k(c_i, a)|}\sum_{Y_t \in \mathcal{Y}_k(c_i, a)}R_t - r(c_i, a) \mid \mathcal{F}^k\right]\right|$$
$$+ \gamma\left|\mathbb{E}\left[\frac{1}{|\mathcal{Y}_k(c_i, a)|}\sum_{Y_t \in \mathcal{Y}_k(c_i, a)}\max_{b \in \mathcal{A}}(\Gamma_{\text{NN}}q^k)(Y_{t+1}, b) \mid \mathcal{F}^k\right] - \mathbb{E}\left[\max_{b \in \mathcal{A}}(\Gamma_{\text{NN}}q^k)(x', b) \mid c_i, a, \mathcal{F}^k\right]\right|.$$

We can bound the first term on the RHS by using Lipschitz continuity of the reward function:

$$\left|\mathbb{E}\left[\frac{1}{|\mathcal{Y}_k(c_i, a)|}\sum_{Y_t \in \mathcal{Y}_k(c_i, a)}R_t - r(c_i, a) \mid \mathcal{F}^k\right]\right|$$
$$= \left|\mathbb{E}\left[\mathbb{E}\left[\frac{1}{|\mathcal{Y}_k(c_i, a)|}\sum_{Y_t \in \mathcal{Y}_k(c_i, a)}(R_t - r(c_i, a)) \mid \mathcal{Y}_k, \mathcal{F}^k\right] \mid \mathcal{F}^k\right]\right|$$
$$= \left|\mathbb{E}\left[\frac{1}{|\mathcal{Y}_k(c_i, a)|}\sum_{Y_t \in \mathcal{Y}_k(c_i, a)}(r(Y_t, a) - r(c_i, a)) \mid \mathcal{F}^k\right]\right|$$
$$\leq \mathbb{E}\left[\frac{1}{|\mathcal{Y}_k(c_i, a)|}\sum_{Y_t \in \mathcal{Y}_k(c_i, a)}|r(Y_t, a) - r(c_i, a)| \mid \mathcal{F}^k\right]$$
$$\leq \mathbb{E}\left[\frac{1}{|\mathcal{Y}_k(c_i, a)|}\sum_{Y_t \in \mathcal{Y}_k(c_i, a)}M_r\rho(Y_t, c_i) \mid \mathcal{F}^k\right] \qquad \text{Lipschitz continuity of } r(\cdot, a)$$
$$\leq M_r h \qquad\qquad\qquad\qquad\qquad\qquad\qquad\qquad\qquad\qquad\qquad \rho(Y_t, c_i) \leq h$$

The second term on the right hand side can be bounded as follows:

$$\left| \mathbb{E}\left[ \frac{1}{|\mathcal{Y}_k(c_i,a)|} \sum_{Y_t \in \mathcal{Y}_k(c_i,a)} \max_{b \in \mathcal{A}}(\Gamma_{\mathrm{NN}}q^k)(Y_{t+1},b) \mid \mathcal{F}^k \right] - \mathbb{E}\left[ \max_{b \in \mathcal{A}}(\Gamma_{\mathrm{NN}}q^k)(x',b) \mid c_i,a,\mathcal{F}^k \right] \right|$$

$$\leq \mathbb{E}\left[ \left| \frac{1}{|\mathcal{Y}_k(c_i,a)|} \sum_{Y_t \in \mathcal{Y}_k(c_i,a)} \int_{\mathcal{X}} \left[ \max_{b \in \mathcal{A}}(\Gamma_{\mathrm{NN}}q^k)(y,b) \right] p(y \mid Y_t,a)\lambda(dy) \right. \right.$$

$$\left. \left. - \int_{\mathcal{X}} \left[ \max_{b \in \mathcal{A}}(\Gamma_{\mathrm{NN}}q^k)(y,b) \right] p(y \mid c_i,a)\lambda(dy) \right| \mid \mathcal{F}^k \right]$$

$$= \mathbb{E}\left[ \left| \frac{1}{|\mathcal{Y}_k(c_i,a)|} \sum_{Y_t \in \mathcal{Y}_k(c_i,a)} \int_{\mathcal{X}} \left[ \max_{b \in \mathcal{A}}(\Gamma_{\mathrm{NN}}q^k)(y,b) \right] (p(y \mid Y_t,a) - p(y \mid c_i,a))\,\lambda(dy) \right| \mid \mathcal{F}^k \right]$$

$$\leq \mathbb{E}\left[ \frac{1}{|\mathcal{Y}_k(c_i,a)|} \sum_{Y_t \in \mathcal{Y}_k(c_i,a)} \int_{\mathcal{X}} \left[ \max_{b \in \mathcal{A}}(\Gamma_{\mathrm{NN}}q^k)(y,b) \right] |p(y \mid Y_t,a) - p(y \mid c_i,a)|\,\lambda(dy) \mid \mathcal{F}^k \right]$$

$$\leq \mathbb{E}\left[ \frac{\sup_{y \in \mathcal{X}} \max_{b \in \mathcal{A}}(\Gamma_{\mathrm{NN}}q^k)(y,b)}{|\mathcal{Y}_k(c_i,a)|} \sum_{Y_t \in \mathcal{Y}_k(c_i,a)} \int_{\mathcal{X}} |p(y \mid Y_t,a) - p(y \mid c_i,a)|\,\lambda(dy) \mid \mathcal{F}^k \right]$$

$$= \sup_{y \in \mathcal{X}} \max_{b \in \mathcal{A}} \left| \sum_{j=1}^{n} K(y,c_j)q^k(c_j,b) \right| \times \mathbb{E}\left[ \frac{1}{|\mathcal{Y}_k(c_i,a)|} \sum_{Y_t \in \mathcal{Y}_k(c_i,a)} \int_{\mathcal{X}} |p(y \mid Y_t,a) - p(y \mid c_i,a)|\,\lambda(dy) \mid \mathcal{F}^k \right]$$

$$\leq \max_{c_j \in \mathcal{X}_h} \max_{b \in \mathcal{A}} \left| q^k(c_j,b) \right| \times \mathbb{E}\left[ \frac{1}{|\mathcal{Y}_k(c_i,a)|} \sum_{Y_t \in \mathcal{Y}_k(c_i,a)} \int_{\mathcal{X}} W_p(y)\rho(Y_t,c_i)\lambda(dy) \mid \mathcal{F}^k \right]$$

$$\leq \left\| q^k \right\|_{\infty} \times \mathbb{E}\left[ \frac{1}{|\mathcal{Y}_k(c_i,a)|} \sum_{Y_t \in \mathcal{Y}_k(c_i,a)} \int_{\mathcal{X}} W_p(y)h\lambda(dy) \mid \mathcal{F}^k \right]$$

$$\leq \left\| q^k \right\|_{\infty} hM_p.$$

Putting together, we have

$$\left| \mathbb{E}\left[ w^{k+1}(c_i,a) \mid \mathcal{F}^k \right] \right| \leq h(M_r + \gamma M_p \left\| q^k \right\|_{\infty}), \quad \forall (c_i,a) \in \mathcal{Z}_h.$$

Hence the noise $w^{k+1}$ satisfies Assumption 1 of Theorem 3 with

$$\delta_1 = hM_r, \delta_2 = h\gamma M_p.$$

From Lemma 3, we have

$$\left| w^{k+1}(c_i,a) - \mathbb{E}\left[ w^{k+1}(c_i,a) \mid \mathcal{F}^k \right] \right| \leq 2V_{\max}, \quad \forall (c_i,a) \in \mathcal{Z}_h,$$

$$\left\| q^k \right\|_{\infty} \leq V_{\max}.$$

Therefore, the remaining Assumptions 2-3 of Theorem 3 are satisfied. And the update algorithm uses the learning rate suggested in Theorem 3. Therefore, we conclude that for each $0 < \varepsilon < 2V_{\max}\beta$ (since $\beta \geq 1$ and hence $2V_{\max}\beta \leq \min\{2V_{\max}\beta, 4V_{\max}\beta^2\}$), after

$$k = \frac{192V_{\max}^3\beta^4}{\varepsilon^3} \log\left( \frac{128N_h |\mathcal{A}| V_{\max}\beta^4}{\delta\varepsilon^2} \right) + \frac{6V_{\max}(\beta - 1)}{\varepsilon}$$

iterations of (9), with probability at least $1 - \delta$, we have

$$\left\| q^k - q_h^* \right\|_{\infty} \leq \beta h(M_r + \gamma M_p V_{\max}) + \varepsilon.$$

$\square$

To prove Theorem 1, we need the following result which bounds the number of time steps required to visit all ball-actions $k$ times with high probability.

**Lemma 6.** *(Lemma 14 in [2], rephrased) Let Assumption 2 hold. Then for all initial state $x_0 \in \mathcal{X}$, and for each integer $k \geq 4$, after a run of $T = 8kL_h \log \frac{1}{\delta}$ steps, the finite space $\mathcal{Z}_h$ is covered at least $k$ times under the policy $\pi$ with probability at least $1 - \delta$ for any $\delta \in (0, \frac{1}{e})$.*

Now we are ready to prove Theorem 1.

*Proof.* We denote by $\tilde{Q}_h^k$ the extension of $q^k$ to $\mathcal{Z}$ via the nearest neighbor operation, i.e., $\tilde{Q}_h^k = \Gamma_{\mathrm{NN}} q^k$. Recall that $Q_h^*$ is the extension of $q_h^*$ (the fixed point of $Gq = q$) to $\mathcal{Z}$. We have

$$
\begin{aligned}
\left\| \tilde{Q}_h^k - Q^* \right\|_\infty &\leq \left\| \tilde{Q}_h^k - Q_h^* \right\|_\infty + \left\| Q_h^* - Q^* \right\|_\infty \\
&= \left\| \Gamma_{\mathrm{NN}} q^k - \Gamma_{\mathrm{NN}} q_h^* \right\|_\infty + \left\| Q_h^* - Q^* \right\|_\infty \\
&\leq \left\| q^k - q_h^* \right\|_\infty + \left\| Q_h^* - Q^* \right\|_\infty \qquad \Gamma_{\mathrm{NN}} \text{ is non-expansive} \\
&\leq \left\| q^k - q_h^* \right\|_\infty + \beta C h \qquad\qquad\quad \text{Lemma 5}
\end{aligned}
$$

It follows from Theorem 4 that, after

$$
k = \frac{192 V_{\max}^3 \beta^4}{\varepsilon_0^3} \log\left( \frac{128 N_h \, |\mathcal{A}| \, V_{\max}^2 \beta^4}{\delta \varepsilon_0^2} \right) + \frac{6 V_{\max}(\beta - 1)}{\varepsilon_0}
$$

iterations, with probability at least $1 - \delta$, we have

$$
\left\| \tilde{Q}_h^k - Q^* \right\|_\infty \leq \beta h (M_r + \gamma M_p V_{\max}) + \beta C h + \varepsilon_0 = 2 \beta C h + \varepsilon_0.
$$

By setting $\varepsilon_0 = \frac{\varepsilon}{2}$ and $h^*(\varepsilon) = \frac{\varepsilon}{4\beta C}$, we have $\left\| \tilde{Q}_h^k - Q^* \right\|_\infty \leq \varepsilon$. Let $N_{h^*}$ be the $h^*$-covering number of the metric space $(\mathcal{X}, \rho)$. Plugging the result of Lemma 6 concludes the proof of Theorem 1. $\qquad\square$

## F   Proof of Corollary 1

*Proof.* Since the probability measure $\nu$ is uniform over $\mathcal{X}$, we have $\nu_{\min} \triangleq \min_{i \in [N_{h^*}]} \nu(\mathcal{B}_i) = O(\frac{1}{N_{h^*}})$. By Proposition 1, the expected covering time of a purely random policy is upper bounded by

$$
L_{h^*} = O\left( \frac{m N_{h^*} |\mathcal{A}|}{\psi} \log(N_{h^*} |\mathcal{A}|) \right).
$$

By Proposition 4.2.12 in [46], the covering number $N_{h^*}$ of $\mathcal{X} = [0, 1]^d$ scales as $O\big((1/h^*)^d\big)$, which is $O\big((\beta/\varepsilon)^d\big)$ with $h^* = \frac{\varepsilon}{4\beta C}$.

From Theorem 1, with probability at least $1 - \delta$ we have $\left\| Q_{h^*}^T - Q^* \right\|_\infty \leq \varepsilon$, after at most

$$
T = O\left( \frac{|\mathcal{A}| \beta^{d+7}}{\varepsilon^{d+3}} \log\left( \frac{2}{\delta} \right) \log\left( \frac{|\mathcal{A}| \beta^d}{\varepsilon^d} \right) \log\left( \frac{|\mathcal{A}| \beta^{d+6}}{\delta \varepsilon^{d+2}} \right) \right)
$$

steps. Corollary 1 follows after absorbing the dependence on $|\mathcal{A}|, d, \beta$ into $\kappa \equiv \kappa(|\mathcal{A}|, d, \beta)$ and doing some algebra. $\qquad\square$

## G   Proof of Theorem 2

We prove Theorem 2 by connecting the problem of estimating the value function in MDPs to the problem of non-parametric regression, and then leveraging known minimax lower bound for the latter. In particular, we show that a class of non-parametric regression problem can be embedded in an MDP problem, so any algorithm for the latter can be used to solve the former. Prior work on non-parametric regression[45, 39] establishes that a certain number of observations is *necessary* to achieve a given accuracy using *any* algorithms, hence leading to a corresponding necessary condition for the sample size of estimating the value function in an MDP problem.

We now provide the details.

**Step 1. Non-parametric regression**

Consider the following non-parametric regression problem: Let $\mathcal{X} := [0,1]^d$ and assume that we have $T$ independent pairs of random variables $(x_1, y_1), \ldots, (x_T, y_T)$ such that

$$\mathbb{E}\left[y_t | x_t\right] = f(x_t), \qquad x_t \in \mathcal{X} \tag{25}$$

where $x_t \sim \text{uniform}(\mathcal{X})$ and $f : \mathcal{X} \to \mathbb{R}$ is the unknown regression function. Suppose that the conditional distribution of $y_t$ given $x_t = x$ is a Bernoulli distribution with mean $f(x)$. We also assume that $f$ is 1-Lipschitz continuous with respect to the Euclidean norm, i.e.,

$$|f(x) - f(x_0)| \leq |x - x_0|, \quad \forall x, x_0 \in \mathcal{X}.$$

Let $\mathcal{F}$ be the collection of all 1-Lipschitz continuous function on $\mathcal{X}$, i.e.,

$$\mathcal{F} = \text{Lip}\left(\mathcal{X}, 1\right) = \left\{h | h \text{ is a 1-Lipschitz function on } \mathcal{X}\right\},$$

where $\text{Lip}(\cdot, \cdot)$ is as defined in Section 2. The goal is to estimate $f$ given the observations $(x_1, y_1), \ldots, (x_T, y_T)$ and the prior knowledge that $f \in \mathcal{F}$.

It is easy to verify that the above problem is a special case of the non-parametric regression problem considered in the work by Stone [39] (in particular, Example 2 therein). Let $\hat{f}_T$ denote an arbitrary (measurable) estimator of $f$ based on the training samples $(x_1, y_1), \ldots, (x_T, y_T)$. By Theorem 1 in [39], we have the following result: there exists a $c > 0$ such that

$$\lim_{T \to \infty} \inf_{\hat{f}_T} \sup_{f \in \mathcal{F}} \Pr\left(\left\|\hat{f}_T - f\right\|_\infty \geq c\left(\frac{\log T}{T}\right)^{\frac{1}{2+d}}\right) = 1, \tag{26}$$

where infimum is over all possible estimators $\hat{f}_T$.

Translating this result to the non-asymptotic regime, we obtain the following theorem.

**Theorem 5.** *Under the above assumptions, for any number $\delta \in (0,1)$, there exits some numbers $c > 0$ and $T_\delta$ such that*

$$\inf_{\hat{f}_n} \sup_{f \in \mathcal{F}} \Pr\left(\left\|\hat{f}_T - f\right\|_\infty \geq c\left(\frac{\log T}{T}\right)^{\frac{1}{2+d}}\right) \geq \delta, \qquad \text{for all } T \geq T_\delta.$$

**Step 2. MDP**

Consider a class of (degenerate) discrete-time discounted MDPs $(\mathcal{X}, \mathcal{A}, p, r, \gamma)$ where

$$\mathcal{X} = [0,1]^d,$$
$$\mathcal{A} \text{ is finite,}$$
$$p(\cdot|x, a) = p(\cdot|x) \text{ is uniform on } \mathcal{X} \text{ for all } x, a,$$
$$r(x, a) = r(x) \text{ for all } a,$$
$$\gamma \in (0, 1).$$

In words, the transition is uniformly at random and independent of the current state and the actions taken, and the expected reward is independent on the action taken but dependent on the current state.

Let $R_t$ be the observed reward at step $t$. We assume that the distribution of $R_t$ given $x_t$ is $\text{Bernoulli}(r(x_t))$, independently of $(x_1, x_2, \ldots, x_{t-1})$. The expected reward function $r(x_t) = \mathbb{E}\left[R(x_t)|x_t\right]$ is assumed to be 1-Lipschitz and bounded.

It is easy to see that for all $x \in \mathcal{X}, a \in \mathcal{A}$,

$$Q^*(x, a) = V^*(x) = r(x) + \gamma \mathbb{E}\left[V^*(X')|x\right]$$
$$= r(x) + \gamma \int_{\mathcal{X}} V^*(y) p(y|x) dy$$
$$= r(x) + \gamma \underbrace{\int_{\mathcal{X}} V^*(y) dy}_{C}, \tag{27}$$

where the last step holds because $p(\cdot|x)$ is uniform. Integrating both sides over $\mathcal{X}$, we obtain

$$C = \int_{\mathcal{X}} r(x)dx + \gamma C,$$

whence

$$C = \frac{1}{1-\gamma} \int_{\mathcal{X}} r(x)dx.$$

It follows from equation (27) that

$$V^*(x) = r(x) + \frac{\gamma}{1-\gamma} \int_{\mathcal{X}} r(y)dy, \qquad \forall x \in \mathcal{X}, \tag{28}$$

and

$$r(x) = V^*(x) - \gamma \int_{\mathcal{X}} V^*(y)dy, \qquad \forall x \in \mathcal{X}. \tag{29}$$

Regardless of the exploration policy used, the sample trajectory $(x_1, x_2, \ldots, x_T)$ is i.i.d. and uniformly random over $\mathcal{X}$, and the observed rewards $(R_1, R_2, \ldots, R_T)$ are independent.

### Step 3. Reduction from regression to MDP

Given a non-parametric regression problem as described in Step 1, we may reduce it to the problem of estimating the value function $V^*$ of the MDP described in Step 2. To do this, we set

$$r(x) = f(x) - \gamma \int_{\mathcal{X}} f(y)dy, \qquad \forall x \in \mathcal{X}$$

and

$$R_t = y_t, \qquad t = 1, 2, \ldots, T.$$

In this case, it follows from equations (28) and (29) that the value function is given by $V^* = f$. Moreover, the expected reward function $r(\cdot)$ is 1-Lipschitz as it is just $f(\cdot)$ minus a constant, so the assumptions of the MDP in Step 2 are satisfied. This reduction shows that the MDP problem is at least as hard as the nonparametric regression problem, so a lower bound for the latter is also a lower bound for the former. Applying Theorem 5 yields the result stated in Theorem 2.