[Reviews · NeurIPS 2018]

Reviewer 1



This work analyzes the sample complexity of a nearest-neighbor Q-learning algorithm. It tackles the case of continuous state space and discrete action space, where discretization of the state space is applied. The paper is well written and relatively easy to follow. The theory and definitions are sound, and the proofs seem to be valid (though I did not verify all of the appendix). I also appreciate the elegancy of the NN Bellman operator in (5). Nonetheless, the work suffers from several weaknesses. Given bellow is a list of remarks regarding these weaknesses and requests for clarifications and updates to the manuscript. - The algorithm’s O(1/(\esiplon^3 (1-\gamma)^7)) complexity is extremely high. Of course, this is not practical. Notice that as opposed to the nice recovery time O(\epsilon^{-(d+3)}) result, which is almost tight, the above complexity stems from the algorithm’s design. - Part of the intractability of the algorithm comes from the requirement of full coverage of all ball-action pairs, per each iteration. This issue is magnified by the fact that the NN effective distance, h^*, is O(\epsilon (1-\gamma)). This implies a huge discretized state set, which adds up to the above problematic complexity. The authors mention (though vaguely) that the analysis is probably loose. I wonder how much of the complexity issues originate from the analysis itself, and how much from the algorithm’s design. - In continuation to the above remark, what do you think can be done (i.e. what minimal assumptions are needed) to relax the need of visiting all ball-action pairs with each iteration? Alternatively, what would happen if you partially cover them? - Table 1 lacks two recent works [1,2] (see below) that analyze the sample complexity of parametrized TD-learning algorithms that has all ‘yes’ values in the columns except for the ‘single sample path’ column. Please update accordingly. - From personal experience, I believe the Lipschitz assumption is crucial to have any guarantees. Also, this is a non-trivial assumption. Please stress it further in the introduction, and/or perhaps in the abstract itself. - There is another work [3] that should also definitely be cited. Please also explain how your work differs from it. - It is written in the abstract and in at least one more location that your O(\epsilon^{-(d+3)}) complexity is tight, but you mention a lower bound which differs by a factor of \epsilon^{-1}. So this is not really tight, right? If so, please rephrase. - p.5, l.181: ‘of’ is written twice. p.7, l.267: ‘the’ is written twice. References: [1] Finite Sample Analyses for TD (0) with Function Approximation, G Dalal, B Szörényi, G Thoppe, S Mannor, AAAI 2018 [2] Finite Sample Analysis of Two-Timescale Stochastic Approximation with Applications to Reinforcement Learning G Dalal, B Szörényi, G Thoppe, S Mannor, COLT 2018 [3] Batch Mode Reinforcement Learning based on the Synthesis of Artificial Trajectories, Raphael Fonteneau, Susan A. Murphy, Louis Wehenkel, and Damien Ernst, Annals of Operations Research 2013

Reviewer 2



The paper provides a finite sample analysis of a reinforcement learning algorithm for continuous state spaces. Given a set of balls covering the compact state space, the RL algorithm learns and stores the optimal Q function for all ball center-action pairs. The Q function is generalized over the whole state-action space using nearest neighbor(s) regression. The main result of the paper is that under some assumptions on the reward and transition, such that Q* is Lipschitz continuous, the proposed algorithm will return a Q function epsilon close to Q* (in sup-norm) after T transition, with T linear in the covering time (expected time for the sampling policy to visit every ball-action pair at least once) and cubic in 1/epsilon. I did not go through all the proofs in the appendix. Otherwise, the paper is well written and very informative. The only limitation I see is that the analyzed algorithm might not be of practical use because of the discretization of the state space and the Lipschitz assumption. The Lipschitz assumption of the reward function might be penalizing since several problems have non continuous reward functions (e.g. positive reward whenever angle is less than some small value); similarly assumption made to the transition function will not work in e.g. robotics whenever there is contact. These assumptions on the reward and transition are not present in related work cited in the paper ([41, 27]), on the other side these papers do not provide a finite sample analysis. As far as I know this constitutes a clear improvement over the state of the art.

Reviewer 3



This paper considers an online Q-learning method using nearest neighbor regression for continuous state space reinforcement learning. The method performs discretization using finite number of center points, and the state-action value function q is online updated using nearest neighbor averaging (regression), followed by the batch update after all states are visited at least once. The authors provide a PAC bound for the estimation of correct value function in this setting. The topic of this paper is fundamental and important for providing a theoretical guarantee of estimating the true value function with a polynomial time complexity. The idea of using nearest neighbor regression is not new (e.g. kernel-based reinforcement learning [3]), but the proposed method uses an online setting which makes this work novel compared with previous methods. Though this work presents an interesting algorithm and its theory, the assumptions for designing the algorithm is strong for real-world applications. The authors increase steps from t to t+1 once all states are visited at least once, but often some states are never reached depending on the starting point. In many systems, people consider the controllability claiming the existence of states that are never reached by any action. In this manner, the derived sufficient number of steps T (between lines 264 and 265) is too big even though it is polynomial on model parameters. (In fact, it is near polynomial in terms of \epsilon) The authors have partly explained these limitations in several places, but the information for practitioners has to clearly appear in the main contents. The authors are recommended to present a simple synthetic and real-world example showing the relationship between the number of steps T and the quality of the learned q function. In particular, the proposed algorithm will be difficult to succeed learning with states in a high dimensional space. Please try to show the algorithm works with high-dimensional states for a simple problem. In terms of the presentation, the authors have to include the algorithm part of the Appendix into the main text. The explanation of the theorem can be improved for the qualitative analysis of the effect of h, \epsilon, and N_h. The authors need to provide the sketch of the proof from the perspective how those parameters are related each other in the proof.